# Ambient carbon dioxide concentration correlates with SARS-CoV-2 aerostability and infection risk

Allen Haddrell [1] ✉, Henry Oswin [1], Mara Otero-Fernandez [1], Joshua F. Robinson [2], Tristan Cogan [3], Robert Alexander [4], Jamie F. S. Mann [3], Darryl Hill [4], Adam Finn [4,5], Andrew D. Davidson [4] ✉ & Jonathan P. Reid [1] ✉

An improved understanding of the underlying physicochemical properties of respiratory aerosol that influence viral infectivity may open new avenues to mitigate the transmission of respiratory diseases such as COVID-19. Previous studies have shown that an increase in the pH of respiratory aerosols following generation due to changes in the gas-particle partitioning of pH buffering bicarbonate ions and carbon dioxide is a significant factor in reducing SARS-CoV-2 infectivity. We show here that a significant increase in SARS-CoV-2 aerostability results from a moderate increase in the atmospheric carbon dioxide concentration (e.g. 800 ppm), an effect that is more marked than that observed for changes in relative humidity. We model the likelihood of COVID-19 transmission on the ambient concentration of $CO_2$, concluding that even this moderate increase in $CO_2$ concentration results in a significant increase in overall risk. These observations confirm the critical importance of ventilation and maintaining low $CO_2$ concentrations in indoor environments for mitigating disease transmission. Moreover, the correlation of increased $CO_2$ concentration with viral aerostability need to be better understood when considering the consequences of increases in ambient $CO_2$ levels in our atmosphere.

The inhalation of respiratory aerosol containing the severe acute respiratory syndrome coronavirus-2 (SARS-CoV-2) has been identified as an important route of transmission in the spread of coronavirus disease 2019 (COVID-19)[1]. As for all respiratory viral infections, a sufficient viral dose must be delivered to the respiratory system of an uninfected individual for disease transmission to occur. For COVID-19, this equates to inhalation of a sufficient quantity of aerosolized/inhalable and infectious SARS-CoV-2 viral particles. The minimal infectious dose is a function of many parameters, such as mucosal

immunity[2], prior infection[3], and immunization status[4]. Regardless of the infectious dose required, the cumulative viral load of the air inhaled will necessarily correlate with the overall risk. Thus, understanding how environmental factors affect the aerosolized viral load over time will contribute to the assessment of the risk of transmission.

At their core, many of the non-pharmaceutical interventions implemented to mitigate the risk of COVID-19 transmission are centered on the removal of infectious aerosolized virus from a given space. The aerosolized viral load may be altered physically by lowering

[1]School of Chemistry, Cantock's Close, University of Bristol, Bristol, UK. [2]Institut für Physik, Johannes Gutenberg-Universität Mainz, Mainz, Germany. [3]Bristol Veterinary School, University of Bristol, Langford House, Langford, Bristol, UK. [4]School of Cellular and Molecular Medicine, University of Bristol, Bristol, UK. [5]School of Population Health Sciences, University of Bristol, Bristol, UK. ✉e-mail: A.Haddrell@bristol.ac.uk; Andrew.Davidson@bristol.ac.uk; J.P.Reid@bristol.ac.uk

the number of virus-containing particles. For example, changing aerosol production rates (e.g., via lowering the volume of singing or talking)[5,6], crowding/social distancing policies[7], mask wearing[8], and improved ventilation[9] all reduce the total number of virus-containing aerosol droplets. The viral load within aerosol droplets may also be altered, so that the infectivity of the viral particles themselves is changed through processes such as UV germicidal irradiation[10] or by adjustments to environmental conditions such as relative humidity (RH)[11] or temperature[12]. In addition to these intentional methods of disinfection, aerosolized viruses are known to lose their infectivity over time, although the precise mechanisms driving this loss remain the subject of much debate[13]. A comprehensive understanding of all these processes/conditions, as well as the interconnections between them, is necessary to facilitate the development of more effective mitigation strategies.

While many of the unique properties of aerosol have been hypothesized to play a role in the loss of viral infectivity, we have reported recently that a high pH (alkaline, pH > 10) in respiratory aerosol surrogates is a significant contributor driving its loss[14,15]. The high pH reached by respiratory aerosol is a consequence of the mucosal liquids from which it originates (e.g., saliva, lung fluid) which contain elevated levels of bicarbonate[16]. Following droplet generation, the pH of the neutral droplet begins to rise as the bicarbonate evaporates from the droplet in the form of gaseous carbon dioxide ($CO_2$) (Eq. 1):

$$H^+_{(aq)} + HCO^-_{3(aq)} \leftrightarrow H_2CO_{3(aq)} \leftrightarrow CO_{2(g)} + H_2O \qquad (1)$$

The maximum pH the aerosol achieves, as well as the time taken to reach it, are unclear and debated[17]. Both are a function of numerous parameters including the RH, initial aerosol droplet size, aerosol equilibrium size, initial bicarbonate concentration, and ambient $CO_2$ concentration ($[CO_{2(g)}]$). What is clear is that measurements of human exhaled aerosol[18–20] and saliva[21] have both shown consistently that exhaled respiratory fluids are significantly more alkaline than the fluids within the respiratory tract from which they originate. Over longer time periods, the high pH of the aerosol may be neutralized through exposure to trace acidic gases, another poorly defined process that requires more investigation[17]. When compared to the vast majority of environmental aerosol, this pH dynamic is a peculiarity of respiratory aerosol and is critical for understanding the aerostability of respiratory viruses.

If, as reported, the pH (alkalinity) of respiratory aerosol is a major driver in the loss of viral infectivity in the aerosol phase, it can be inferred that $[CO_{2(g)}]$ has an effect on the aerostability of SARS-CoV-2 via the equilibrium described in Eq. 1. This raises three questions: over what ambient concentration range does $CO_{2(g)}$ impact infectivity, to what degree is the viral infectivity decay profile affected by $CO_{2(g)}$, and how does this change in infectivity affect overall risk of disease transmission?

Since 2020, $CO_2$ monitors have become commonly used as an indicator of potential risk of SARS-CoV-2 transmission as the $[CO_{2(g)}]$ serves as a proxy for overall ventilation efficiency and, thus, a predictor of total aerosol viral load. The source of both the aerosolized virus and $CO_2$ are the same (exhalation) and both are reduced through standard mitigation techniques such as ventilation[22], justifying the use of $[CO_{2(g)}]$ as a proxy indicator of transmission risk. We investigate here if elevated ambient levels of $[CO_{2(g)}]$ add a further factor to an increased transmission risk by altering the aerostability of SARS-CoV-2. This is accomplished using the Controlled Electrodynamic Levitation and Extraction of Bioaerosol onto a Substrate (CELEBS)[14,15,23–26] to systematically explore the effects of environmental factors such as ambient $[CO_{2(g)}]$ and RH on the aerostability of the SARS-CoV-2 Delta and Omicron variants of concern (VOC). The CELEBS is a next-generation aerobiology technique that allows studies of changes in the viral infectivity within a small population of levitated droplets of a near

identical size and composition as a function of time, temperature, relative humidity, and complete gas-phase composition. Additionally, viral decay from <5 s to hours can be readily measured with this technique. Finally, we estimate the effect of changes in the ambient $[CO_{2(g)}]$ on the risk of transmission using the established Wells-Riley model.

## Results

### The BA.2 omicron VOC is more aerostable than the delta VOC

Previously, we reported that the aerostability of the VOCs of SARS-CoV-2 (wild type, Alpha, Beta, and Delta) correlated with the variant's stability in an alkaline growth medium over short-time periods (under five minutes)[15]. Specifically, it was reported that as the virus has evolved from wild type through to Delta VOC, it has become both more sensitive to high pH and less aerostable. The relationship between pH and the aerostability of the (Omicron) BA.2 VOC is compared to the Delta VOC to further explore this comparison (Fig. 1). Firstly, the aerostabilities of the Delta and BA.2 VOCs within aerosol droplets are reported at moderate/low (40%) and high RH (90%) values in Fig. 1a. At 40% RH, the Delta and BA.2 VOCs exhibit similar decay profiles. This is consistent with our previous studies where all VOCs exhibited a similar decay profile when the RH is below the droplet efflorescence threshold, highlighted by the near-instantaneous loss of ~50% of viral infectivity associated with the efflorescence event. At 90% RH, the overall rate of decay of the BA.2 VOC is much slower than the Delta VOC. At 5 min, relative to the Delta VOC, the total viable aerosolized viral load of the BA.2 VOC is 1.7 times higher. Excluding data points in the transient decay (<1 min) to focus on the more gradual decay from 2 min onwards, our analysis reveals significant differences in infectivity between the Delta and BA.2 VOCs. At the 2 min time point, we observe a statistically significant difference in the infectivity of the Delta and Omicron VOCs of $26 \pm 6$ % ($p = 3 \times 10^{-4}$, with 19.8 effective degrees of freedom via Satterthwaite approximation) at 90% RH. This effect persists at the 5 min mark, with a difference of $19 \pm 8$ % ($p = 0.03$, with 18.9 degrees of freedom). At 90% RH, the general structures of the decay profiles for both VOCs are consistent with previous VOCs, with an initial lag period of ~15 s, followed by a rapid loss to ~2 min, and a more gradual subsequent decay.

Under high alkaline conditions in the bulk phase, the BA.2 VOC is found to be more resistant to high pH conditions than the Delta VOC assessed via two different measurements of infectivity (Fig. 1b, c). This is consistent with the hypothesis that the differences between aerostability of the Delta and BA.2 VOCs are likely a consequence of their relative stability in a highly alkaline solution. BA.2 is the first VOC of SARS-CoV-2 that we have demonstrated to have an increase in stability at high pH when compared to a prior VOC. The microbiological mechanisms underlying these differences in pH sensitivity remain unclear and are in need of further research.

Collectively, the data shown in Fig. 1 support the hypothesis that high pH achieved in aerosol, with an initial composition that has high abundance of bicarbonate (such as saliva[27] and growth medium), is a major factor in driving loss of viral infectivity in the aerosol phase[15]. The implication of this proposed mechanism is that any gaseous species (e.g., $CO_2$) that can affect aerosol pH is likely to impact viral infectivity.

### SARS-CoV-2 aerostability correlates with the ambient concentration of gas-phase carbon dioxide

In poorly ventilated, occupied, indoor spaces, ambient $[CO_{2(g)}]$ commonly reaches concentrations exceeding 2000 ppm[28] and can reach levels upwards of >5000 ppm in more crowded environments[29]. The impact of elevated $CO_{2(g)}$ levels on the aerostability of SARS-CoV-2 is explored (Fig. 2). Both the sensitivity and the throughput of the CELEBS technique are dependent on the initial viral load of the individual droplets. The maximum titer of the BA.2 VOC that can be grown in the

cell culture is approximately an order of magnitude less than that of the Delta VOC. Thus, in order to explore the effect that $[CO_{2(g)}]$ has on viral aerostability across a broad range of conditions and over long-

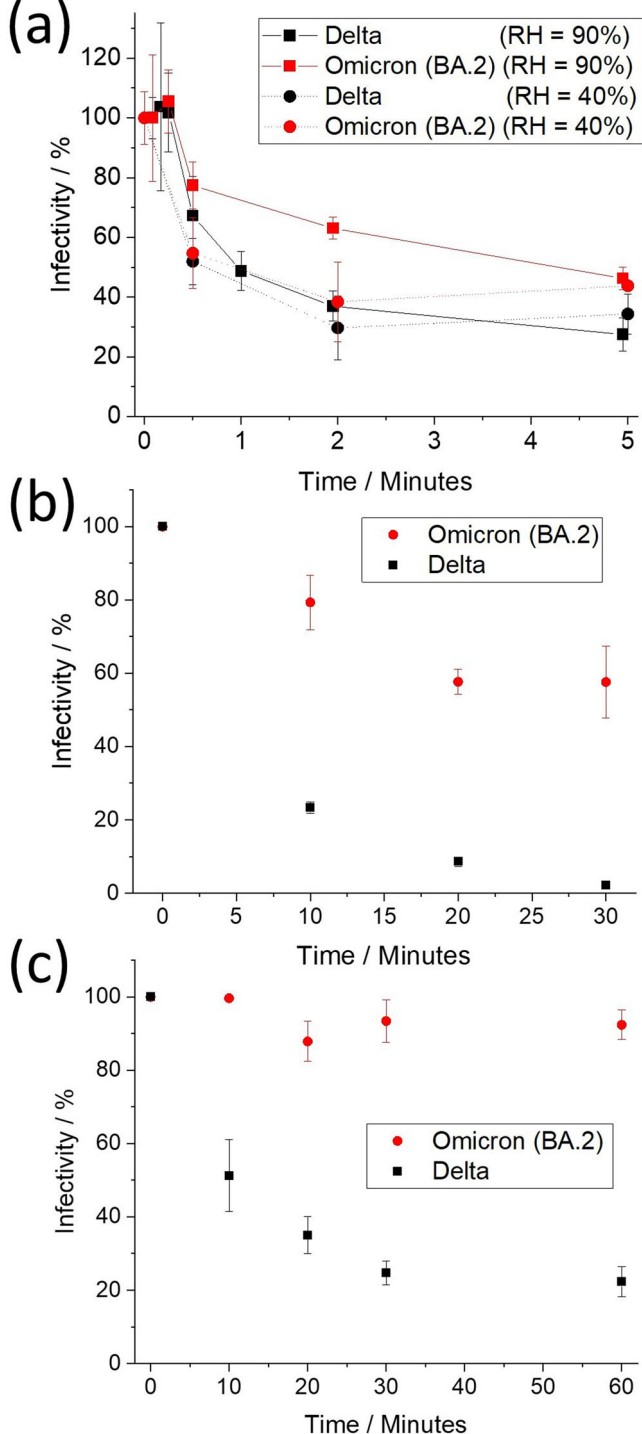

**Fig. 1 | Infectivity over time of the Delta and Omicron (BA.2) VOCs in aerosol and bulk solutions of high pH. a** Infectivity of the Delta and Omicron BA.2 VOCs that have been levitated at RHs of 40% and 90%. Data for 90% at times over 100 s are offset by 5 s to facilitate reader interpretation. **b**, **c** Infectivity of BA.2 and Delta VOCs, respectively, in DMEM 2% FBS bulk solution with pH maintained at 11 and measured by **b** cytopathy and **c** immunostaining. Values are means ± SE. Both **b** and **c** fit a first-order decay; from a *t*-test to compare slopes of regression lines (two-sided), the linear fits of ln(infectivity) vs. time for the Omicron and Delta VOCs are consistent with decay rates that are significant in their difference $p = 0.0002$ ($n = 24$ (independent samples)) for (**b**), $p = 0.027$ ($n = 30$ (independent samples)) for (**c**). Source data are provided as a Source Data file.

time periods, the majority of the testing was conducted with the Delta VOC as it afforded a much higher measurement throughput.

When compared to a typical atmospheric $[CO_{2(g)}]$ (~500 ppm), increasing the $[CO_{2(g)}]$ to just 800 ppm results in a significant increase in viral aerostability after 2 min (Delta VOC, Fig. 2a). No significant difference in infectivity is observed between 800 ppm and 6500 ppm. It is notable that, according to the UK Scientific Advisory Group for Emergencies (SAGE), 800 ppm $[CO_{2(g)}]$ has been identified as the level below which a room is determined to be well-ventilated. When ambient airflow into the CELEBS is substituted with synthetic air (0 ppm $[CO_{2(g)}]$), no change in virus aerostability is observed.

Increasing the $[CO_{2(g)}]$ results in a significant increase in the aerostability of both the Beta and Omicron BA.2 VOCs at 120 s. This suggests that the viral aerostability is dependent on $CO_{2(g)}$ concentration for all SARS-CoV-2 variants. The increase in infectivity of the Omicron variant due to the elevated $[CO_2]$ whilst similar, is slightly lower (+11.7%) than the Beta (+23.4%) and Delta (+36.8%) VOCs. This is likely a product of the differing pH sensitivities for different variants, with the more pH-sensitive variants more sensitive to changes in $[CO_2]$. It is unlikely a similar increase across variants will occur at all time periods for all variants.

The rate of viral infectivity loss correlates with aerosol alkalinity. Elevated $[CO_{2(g)}]$ limits the amount of bicarbonate leaving the droplet (Eq. 1) and, thus, limits the maximum pH that the droplet will reach. A significant improvement in aerostability of SARS-CoV-2 resulting from elevated $[CO_{2(g)}]$ would be expected to increase over time as the droplet will spend less time at the elevated pH[30]. The effect of droplet exposure to elevated $[CO_{2(g)}]$ over prolonged time periods on the infectious viral load is reported in Fig. 2b and it can be seen that elevated $[CO_{2(g)}]$ had a considerable effect on the overall decay profile. Consider first the characteristics of the decay profile of the wild type SARS-CoV-2 in the aerosol phase above the efflorescence point[14]. From droplet generation until ~2 min, no loss of infectivity is observed. After 2 min there is a rapid loss of infectivity over a moderate time period (minutes), followed by a slower decay (tens of minutes). The profile for the Delta VOC is similar but with the initial lag period shortened to ~15 s. In this case, when the $[CO_{2(g)}]$ is elevated, the period of rapid decay is absent or greatly abbreviated and the decay profile transitions directly from lag to a slow decay. As a result, the Delta VOC in elevated $[CO_{2(g)}]$ is as aerostable as the wild type at 500 ppm $CO_2$ after 5 min, resulting in a larger fraction remaining infectious after 20 min ($p = 0.04$). Indeed, elevated $[CO_{2(g)}]$ has a dramatic effect on the remaining relative infectivity of SARS-CoV-2 over time (Fig. 2c). A hypothesis test regression of slope for the data in Fig. 2c found a $p = 0.00038$, consistent with a lower rate of viral decay at elevated $[CO_{2(g)}]$. After 40 min, approximately an order of magnitude more viral infectious particles remain viable in the aerosol phase at elevated $[CO_{2(g)}]$ when compared to the loss expected under ambient (well-ventilated) conditions. This increase in the relative abundance of infectious particles is likely to result in increased risk of transmission of the infection.

Viral aerostability is often reported as having a half-life[31] with the decay assumed to follow first-order (exponential) reaction kinetics. This assumption presupposes that the mechanisms involved in infectivity loss do not change over time, even though the chemical composition and physical conditions inside an aerosol droplet vary over time. The appropriateness of making such assumptions is explored in Fig. 2d. In the bulk phase, the pH-driven decay follows first-order kinetics. This is consistent with high $[OH^-]$ driving the loss of viral infectivity, with this concentration remaining constant over time. However, the decay dynamics are markedly different in the aerosol phase with the rate of loss slowing over time, and slowing more so at higher $[CO_{2(g)}]$. This is consistent with the hypothesis that the aerosol achieves high pH before being buffered towards a neutral pH by trace acidic vapor over longer time periods (condensable carbonic acid in this case), regardless of RH. However, the decay rate is never found to

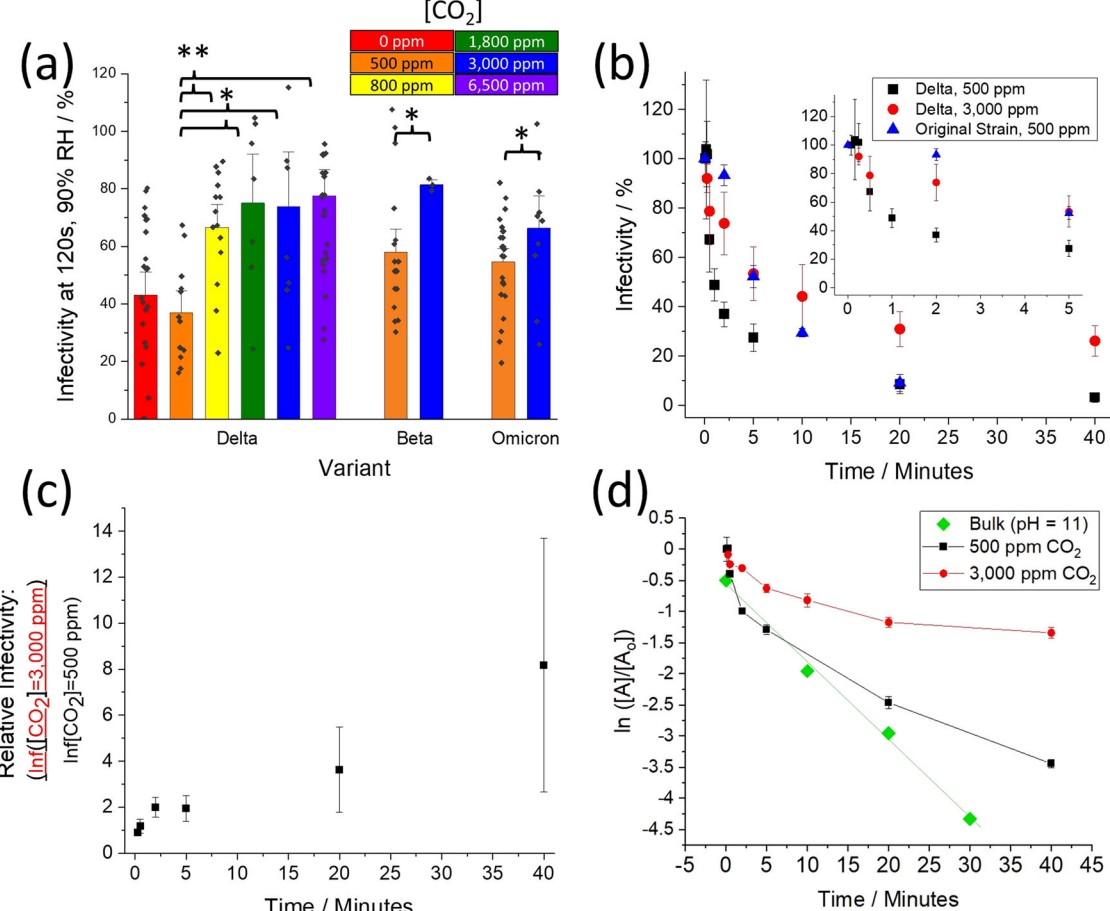

**Fig. 2 | Exploring the effect that $[CO_{2(g)}]$ has on the aerostability of the Delta VOC as measured with the CELEBS. a** Infectivity of the Delta, Beta, and Omicron BA.2 VOCs as a function of ambient concentrations of $CO_{2(g)}$ at 90% RH, 120 s. Statistical significance was assessed using a one-sided, two-sample equal variance, *t*-test (*$p \leq 0.05$, **$p \leq 0.005$, $n = 146$ (independent samples)). Specifically, the significance (*p*-value) between 500 ppm and 800 ppm, 1800 ppm, 3000 ppm, and 6500 ppm was 0.003, 0.025, 0.032, and 0.001, respectively. **b** The effect that an elevated concentration of CO2 has on the decay profile of the Delta VOC and original strain of SARS-CoV-2 at 90% RH. Inset is simply a zoom-in of the first 5 min of the x-axis. Elevating the $[CO_{2(g)}]$ results in a significant difference in overall decay assessed using a one-sided, two-sample equal variance, *t*-test ($n = 188$ (independent

samples)) of the Delta VOC from 2 min onward, where the significance (*p*-value) was 0.007, 0.027, 0.020 and 0.005 for 2, 5, 10 and 40 min, respectively. **c** Relative infectivity of aerosolized Delta VOC exposed to increased $[CO_{2(g)}]$ as a function of time. The ratios were estimated using the raw data in Fig. 2b at time points where the infectivity was measured for both [CO2(g)]. Note that the error bars increase with time results from the infectivity of the "Delta, 500 ppm" data set approaching zero which causes the relative standard deviation to increase. **d** Infectivity of the Delta VOC ($\ln([A]/[A_o])$) as a function of time in aerosol and in MEM at pH = 11 (bulk data offset by −0.5). Least square fit through the Bulk data ($R^2 = 0.995$). Values are means ± SE. Source data are provided as a Source Data file.

increase over the entire time period studied, suggesting that the pH of the aerosol does not pass through neutral to become acidic during the time period when more than 95% of the viral infectivity is lost. Again, this is consistent with the condensation of a weak acid, carbonic acid in this case. However, a half-life of ~80 min could be estimated from the data falling between 20 and 40 min at $[CO_{2(g)}]$ of 3000 ppm if one were to assume a first-order decay. This aligns with the half-lives reported using other measurement approaches in which the $[CO_{2(g)}]$ is neither measured nor controlled[31].

Collectively, the data shown in Fig. 2 show that the interplay between aerosol alkalinity and $CO_2$ has a profound effect on the overall aerostability of SARS-CoV-2. Any increase in $[CO_{2(g)}]$ results in an increase in aerostability.

### Depending on variant pH sensitivity, ambient [CO2] and solute composition can affect viral aerostability more than relative humidity

During the COVID-19 pandemic, many infections were traced to superspreader events[32], suggesting that transmission of the virus over longer distances was possible under some (as yet uncertain)

conditions[1,33]. Conversely, the apparent effectiveness of mitigation strategies such as social distancing regulations[7], use of face shields/masks[34], and installation of plexiglass shields[35] suggests that SARS-CoV-2 was also commonly transmitted over short distances. Thus, it is important to understand how environmental factors affect the aerostability of SARS-CoV-2 over time periods as short as 15 s.

Fixing the time in the aerosol phase at 15 s, the BA.2 VOC is found to be more aerostable than the Delta VOC across a broad range of RH (Fig. 3a). Effectively, during the first 15 s post-aerosol generation there is no loss of infectivity of the BA.2 VOC when the RH is above the efflorescence point of the particle (RH~50%). Below that, the characteristic rapid loss of approximately half of the viral infectivity is observed, a consequence of the efflorescence event. This is consistent with the loss observed for the other SARS-CoV-2 VOCs we have studied[14,15,36] as well as for mouse hepatitis virus (MHV), another coronavirus[37]. At an RH between ~50% and 80%, the Delta VOC is less aerostable than the other VOCs studied, rapidly losing over half of its infectivity within 15 s of aerosolization. Collectively, the data in Fig. 3a show that the initial decay of the Delta VOC is largely RH independent, while the initial decay of the BA.2 VOC is highly RH-dependent.

Given that the BA.2 VOC is more robust than the Delta VOC over the 15 s time period, the impact of $[CO_{2(g)}]$ and [NaCl] on the aerostability of the Delta VOC have been explored further. The effect that a moderate increase in $[CO_{2(g)}]$ has on the aerostability of this SARS-CoV-2 VOC is reported in Fig. 3b. Regardless of RH, increasing the $[CO_{2(g)}]$ will drive the pH of an alkaline respiratory droplet towards neutral to some degree. As a result, at an RH of 80% and below, moderate increases in the $[CO_{2(g)}]$ are shown to increase viral aerostability. This increase in $[CO_{2(g)}]$ results in a doubling of the remaining aerosolized viral load after 15 s for all RH < 80%. Pooling the three datasets for RH < 80%, we verified these differences were statistically significant by performing a two-sample $t$-test ($p = 5 \times 10^{-4}$, with 27.5 effective degrees of freedom). At 90% RH, no viral decay was measured (Fig. 3a, b); droplets injected into an RH of 90% are still evaporating at 15 s and, thus, the conditions in the droplet are not sufficiently different to affect viral infectivity.

## Risk of transmission is highly affected by ambient concentrations of CO2

The dependence of the overall risk of SARS-CoV-2 transmission on the explicit decay dynamics inferred from measurements with the CELEBS technique has been investigated using a Wells-Riley model[38]. Specifically, the impact of environmental factors such as $[CO_{2(g)}]$ and humidity on the likelihood of disease transmission have been explored. The Wells-Riley model is based on transmitted quanta that inherently assume a uniformity in a mixed room, only strictly true for small particles in the bronchiolar and laryngeal modes, both <5 μm diameter. The decay data measured in this study are for droplet sizes in the oral mode (initially >50 μm diameter). We use the infectivity decay data from these large droplet measurements to inform estimates of transmission risk for the small aerosol fraction and estimate the relative changes in risk that result from changes in $[CO_{2(g)}]$. It should be noted that the rate of pH reduction will be dependent on both droplet size and total acid content in the air, where smaller droplets will be neutralized at a faster rate. The effect of size was explored in this study to a limited degree and was found to have a minimal effect within the size range explored (Supplementary Fig. 6). The precise degree to which the droplet size affects the rate of neutralization (and subsequent increased viral aerostability) should be measured in the future.

Typically, models assume that the aerosolized viral decay has a half-life of 1.1 h[39] when estimating the risk of COVID-19 transmission, a rate of decay which is negligible when compared to the effects of even the poorest ventilation. As shown in Fig. 1, viral decay dynamics are more complex than the assumed single exponential decay. To be clear, the rapid early decay in infectivity of aerosolized SARS-CoV-2 we report here, as well as previously[14], does not contradict the consensus opinion that airborne transmission prevails as the dominant mode of transmission. Our objective here is to demonstrate that the decay dynamics reported in Fig. 1a, b are actually consistent with this consensus, especially in indoor environments. We therefore focus on the limit of a well-mixed indoor environment using the Wells-Riley framework and using our refined characterization of the infectivity decay rate.

Central to the Wells-Riley approach is the number of infectious units ("quanta") that remain active in a room. In a ventilated environment, the probability that an aerosolized unit remains in the room after some time is described by Equation 2.

$$p_{active} = 1 - \exp\left(\frac{-t}{\tau_{vent}}\right) \qquad (2)$$

where $\tau_{vent}$ is the characteristic time to cycle air in the room. We consider typical ventilation rates for $\tau_{vent}^{-1}$ in the range 0.5–8 h$^{-1}$ with smaller numbers indicating a poorly ventilated space[40]. Any effect of droplet removal by deposition which may occur for coarser droplets is ignored. The fraction of droplets remaining viable to initiate infection

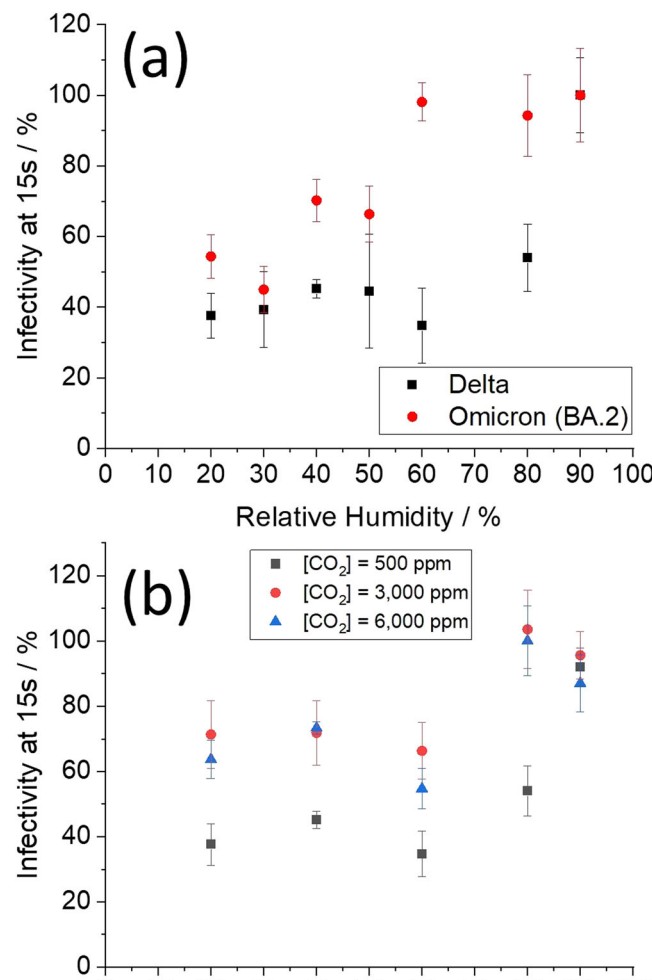

**Fig. 3 | Infectivity of the Delta and Omicron VOCs across a range of RH and $CO_{2(g)}$ concentrations. a** Infectivity of the Delta and Omicron BA.2 variants at 15 s in ambient air as a function of relative humidity. **b** Infectivity of the Delta variant at 15 s as a function of relative humidity and $[CO_{2(g)}]$. Values are means ± SE. A two-way ANOVA of the data in (**b**) indicated that both $CO_{2(g)}$ and RH are significant factors in predicting infectivity, but do not interact, meaning the effect of $CO_{2(g)}$ is similar at different RHs (all 80% and below). $P$-values: RH 0.016, $CO_{2(g)}$ < 0.0001, $n = 169$ (independent samples). Source data are provided as a Source Data file.

is $p_{active}I$, where $I$ is their infectivity. Initially, we consider a fully recirculating ventilation system where the $[CO_{2(g)}]$ remains constant so that we can assume the decay in infectivity directly follows that reported from the CELEBS data; this set-up models a closed Heating, Ventilation and Air-Condition (HVAC) system. We assume the HVAC system perfectly filters the air of aerosol droplets, although this is a crude oversimplification[41]. Later we will allow for varying $[CO_{2(g)}]$ in order to model ventilation and mixing with an outdoor air source (from e.g., opening a window). For convenience, we fit the CELEBS data (Fig. 1a, b) with two exponentially decaying functions (details in Supplementary). In poorly ventilated environments, the viability of aerosolized virus is dominated by the intrinsic decay in infectivity (Fig. 4a). Air recirculation dominates in better-conditioned environments leading to convergence of the long-time decay (Fig. 4b). We compare these model predictions with a simple exponentially decaying infectivity with the widely assumed aerosolized half-life of 1.1 h[31] from drum data.

In the Wells-Riley model, the infection probability $p_I$ is assumed to depend exponentially on the number of infectious units ("quanta")

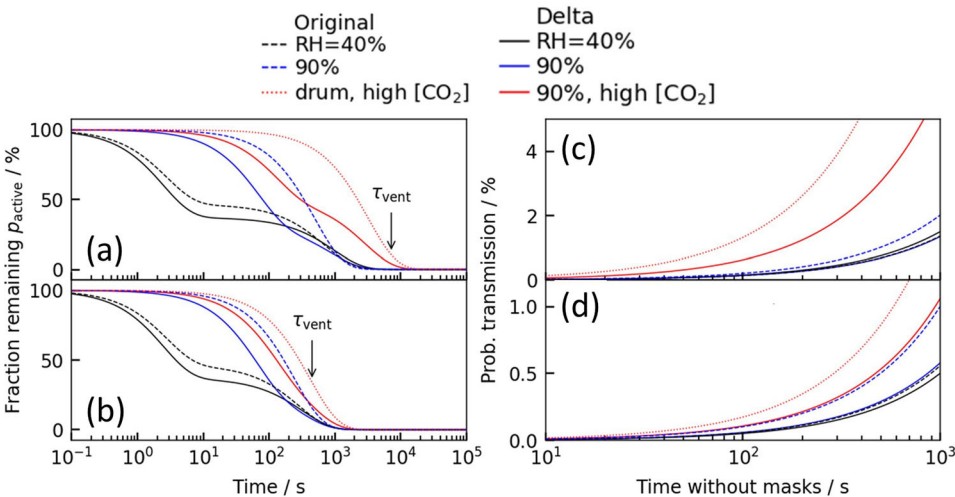

**Fig. 4 | Numerical modeling of risk of indoor transmission in a 300 m³ classroom that combines fits of infectivity data from Figs. 1–3 with the Wells-Riley model for airborne transmission of the Delta VOC in a well-mixed environment.** Fraction of infectious aerosol particles remaining in the **a** poorly ventilated (0.5 air changes per hour) and **b** well-ventilated (8 air changes per hour) classroom where "Time" is the time following exhalation of the infectious aerosol. Probability of onward transmission to a susceptible individual assuming a well-mixed **c** poorly ventilated (0.5 air changes per hour) and **d** well-ventilated (8 air changes per hour) environment, where "Time" is the time following occupation of the room. An infectious quanta production rate of 0.1s-1 is assumed (38). Decay profile in (**a**, **b**) is used to model the transmission risk in (**c**, **d**).

received $n$, i.e. $p_I(t) = 1 - \exp(-n(t))$. This exponential dose-response relationship is essentially a consequence of the independent action hypothesis[42]. The number of quanta $n$ describes the number of infectious viral doses received, incorporating the effects of viral viability and ventilation. The typical number of quanta received increases linearly in time $t$ as $n(t) = c\dot{V}t$ where $c$ is the concentration of quanta in the well-mixed air and $\dot{V}$ is the minute volume of exhaled air (from breathing) which we take to be 7.5 L/min. The steady-state quanta concentration is found where $p_{active}I$ balances the rate that infectious quanta are produced and released into the environment (details in Supplementary). For illustration purposes, we consider a $10 \times 10 \times 3 = 300$ m³ classroom with up to 40 occupants where there is a single infected individual. The quanta production rate by an infected individual is considered to be in the range 0.01–0.1 s⁻¹ for SARS-CoV-2[39]. We assume a value of 0.1 s⁻¹ for illustrative purposes, with the understanding that this factor remains a major source of uncertainty in transmission models. The probability of onward transmission in a poorly ventilated classroom is similar for all datasets at low $[CO_{2(g)}]$ (Fig. 4c, blue and black lines) because only the long-time behavior matters in this well-mixed limit. By contrast, the probability of onward transmission rises much more rapidly for the high $[CO_{2(g)}]$ (Fig 4, red lines). The amplifying effect of $[CO_{2(g)}]$ is visible but less pronounced in a space with good ventilation using, e.g., an open window (Fig. 4d). The probability of transmission is sensitive to parameters with large uncertainties like the quanta production rate, so the absolute values of transmission probability in Fig. 4c, d should not be taken literally. Rather, the important point is that the relative difference in probability between low $[CO_{2(g)}]$ and high $[CO_{2(g)}]$ can be striking; even with good ventilation (Fig. 4d), the probability of onward transmission approximately doubles for $[CO_{2(g)}] = 3000$ ppm over the $[CO_{2(g)}] = 500$ ppm after ~15 min of exposure.

For risk management, we must consider the risk that *any* susceptible individual becomes infected, rather than just a single individual. The probability that at least one individual becomes infected is:

$$1 - \left(1 - p_I(t)\right)^{N-1} \quad (3)$$

where $N$ is the room occupancy. As a measure of risk, we invert this relationship to determine the length of time the classroom space can be shared until there is a 50% chance that secondary transmission has occurred. The probability of a successful transmission (assuming a well-mixed environment) as a function of viral aerostability and ventilation is explored in Fig. 5. Estimates for the Delta variant at high $[CO_{2(g)}]$ are comparable in magnitude to predictions with a decay time assumed from the drum studies. By contrast, the effect of increased $[CO_{2(g)}]$ has a profound effect on the overall risk of transmission. Even in well-ventilated classrooms with 10 air changes per hour (in a recirculating system), we see that the time before an expected transmission occurs is approximately halved by raising the $[CO_{2(g)}]$.

Sustained $[CO_{2(g)}]$ at a higher concentration at a fixed ventilation rate (e.g., more recirculation of room air and less mixing of fresh air) implies lower aerosol pH, greater survival, and shorter time until 50% transmission. Assuming a slower decay rate consistent with the drum data does not incorporate the rapid initial loss of infectivity which means, at the same quanta emission rate, there is more infectious virus and shorter time to reach 50% infectivity. Outdoor air lowers the $CO_2$ concentration, ensuring the pH remains higher, leading to lower infectivity and longer time to reach same infectivity as ACH goes up.

Despite the rapid initial decay (Fig. 1a, b), the Wells-Riley model prediction estimates that long-distance transmission is possible. However, the Wells-Riley model neglects the short-time decay indicating that short-range airborne transmission route (from e.g., direct conversation) may be underestimated in these conventional approaches. Collectively, the risk estimations from the Wells-Riley model demonstrate the importance of ventilation in mitigating risk as it addresses aerosolized viral load on two fronts: firstly, the rate of loss of viral infectivity in the aerosol phase (e.g., lower $[CO_{2(g)}]$ increases decay rate) and, secondly, the physical reduction of the number of viral containing particles (e.g., displaced from the room). The Wells-Riley model demonstrates the importance of $[CO_{2(g)}]$ and RH on long-distance transmission risk. In the future, the effect the rapid loss of infectivity in the aerosol phase at low RH has on short-distance transmission risk should be explored using a CFD model.

## Discussion

In the absence of infectious virus sampling, ambient $[CO_{2(g)}]$ has been shown to indicate increased COVID-19 infection risk through a reduction in effective ventilation and an increase in infectious particle

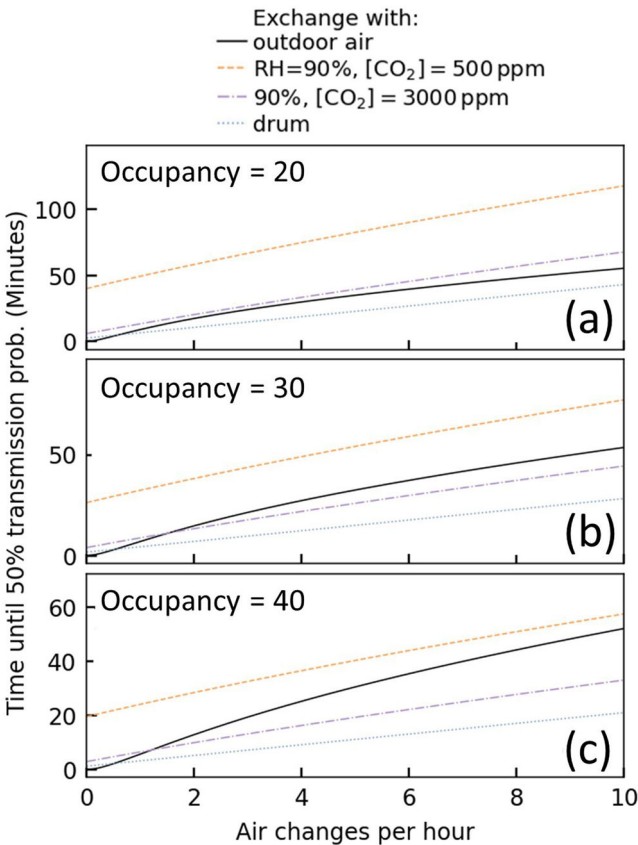

**Fig. 5 | Time until there is a 50% chance that at least one susceptible person will have become infected in an occupied classroom containing a single infected individual, where the viral decay rate is dictated by the RH and [CO$_{2(g)}$] in the replacement air.** We combine estimates of the [CO$_{2(g)}$] in the room with interpolations of the datasets for Delta variant's infectivity at 90% RH to estimate the risk when exchanging with outdoor air (black, 40% RH, 450 ppm [CO$_{2(g)}$]). This interpolation is purely intended to illustrate the nonlinear role of ventilation qualitatively, and so these numbers should not be taken literally. For comparison we also show the expectation at fixed low (purple) and high (orange) [CO$_{2(g)}$] without any interpolation; this models recirculation flow within a HVAC system. The number of occupants is varied **a** 20, **b** 30, and **c** 40. An infectious quanta production rate of 0.1s-1 is assumed.

concentrations[43]. The data presented here suggests that [CO$_{2(g)}$] concentration may be more than just an indicator of poor ventilation or air filtration efficiency. Ultimately the aerosolized virus and CO$_2$ have the same origin, and their mutual interaction increases the overall risk (Fig. 5). This means that the utility of [CO$_{2(g)}$] as a proxy for transmission is of increased value as increased [CO$_{2(g)}$] itself may increase the likelihood of successful transmission by increasing viral viability.

## Broader implications

The ability of [CO$_{2(g)}$] to affect virus aerostability may have broader implications beyond disease transmission and prompts many new avenues of research. For example, the aerostability of a virus being affected by the [CO$_{2(g)}$] raises many questions regarding the potential effect that increases in the [CO$_{2(g)}$] in the atmosphere has on both the transmissibility of extant viruses, as well as on the emergence of novel viruses. [CO$_{2(g)}$] has increased from the preindustrial revolution (275 ppm), through to now (~400 ppm), and may reach upwards of >1000 ppm by the turn of the century[44]. This increase may be enough to improve viral transmission through both increasing the aerostability of the virus outdoors (Fig. 2a), but also increasing the baseline [CO$_{2(g)}$] indoors as well. The degree to which [CO$_{2(g)}$] plays a role in disease

transmission via changes in aerostability specifically needs to be explored further across a range of conditions and particle types.

Respiratory viral infections, such as influenza and rotavirus, are notable as they have a seasonality[45]. There are numerous hypotheses as to what drives this process including, for example, that the dry indoor air over winter may have an effect[46,47]. The effect of RH may also be important with regards to processes such as mucosal immunity[2,48] and plume dynamics[49], and less so for viral infectivity. Seasonal variation in indoor [CO$_{2(g)}$] occurs globally, across a broad range of geographies[50–52]. From the experimental and model data reported here, we hypothesize that the seasonality of respiratory viral infections at the population level may be affected by indoor [CO$_{2(g)}$] as well as changes in RH. Further study is needed to explore this relationship across a broad range of respiratory viruses within respiratory particles of various sizes and compositions.

Moderate increases in [CO$_{2(g)}$] (from 500 to 800 ppm) affecting the aerostability of SARS-CoV-2 have very broad implications with regard to how all previously published aerovirology experiments should be interpreted. Standard experiments involve the nebulization of a virus-containing starting formulation into a confined volume where the aerosol is suspended. The starting formulation will contain some level of bicarbonate, either from the growth medium in which the virus is made, or because the starting formulation is some form of respiratory fluid. If the starting formulation does not have bicarbonate, the utility of the data is questionable as the starting formulation is missing a critical component that is both driving the loss of viral infectivity in respiratory aerosol and necessarily a part of the respiratory system of mammals. The presence of bicarbonate in the starting formulation necessitates that CO$_2$ must be produced during the nebulization process, and there is no physical means to separate the aerosol condensed phase from the gas phase. The amount of CO$_2$ produced will depend on the amount of sample nebulized and has been reported to be greater than the 800 ppm to improve viral aerostability[14]. It is notable that of the five publications that have reported the decay rate of the original variant of SARS-CoV-2[11,31,53–55], only two provided enough information to estimate the total mass of the sample nebulized, while none reported the [CO$_{2(g)}$]. Accordingly, in the study of aerosolized viruses, a major parameter that affects viral aerostability (CO$_2$) is simultaneously intrinsic in the experiment, and not considered.

The decay rate of SARS-CoV-2 in aerosol as measured with a rotating drum, or similar closed system in which the sample is nebulized and the subsequent plume captured, is reported as a half-life of ~1.1 h[10,11,31]. This matches with the half-life reported in this study when the [CO$_{2(g)}$] was elevated (Fig. 2d, after 20 min). Direct confirmation of this is not possible as the [CO$_{2(g)}$] in rotating drum studies has never been reported. In the absence of elevated [CO$_{2(g)}$], the half-life as measured with the CELEBS for any of the SARS-CoV-2 VOCs is <20 min. The ability for moderate increases in [CO$_{2(g)}$] to dramatically increase the aerostability of SARS-CoV-2 (Fig. 2a) largely explains the discrepancy between the aerostability reported using the CELEBS-system with those reported previously using nebulized-based instruments.

## SARS-CoV-2 has a triphasic decay profile

The high time resolution of the CELEBS technology has afforded unique insights into the decay dynamics of aerosolized viruses that have been previously impossible. Historically, the decay of an aerosolized virus has been described as having a half-life. As shown here and in previous studies[14,15,26], this is clearly not the case (Fig. 2d). Rather, the decay profile is a complex process that is highly dependent on both microbiology and aerosol dynamics. From our studies using this next-generation technology, we have found that there are 3 distinct phases of aerosolized viral decay: the Lag Phase, the Dynamic Phase, and the Slow Decay Phase as shown schematically in Fig. 6.

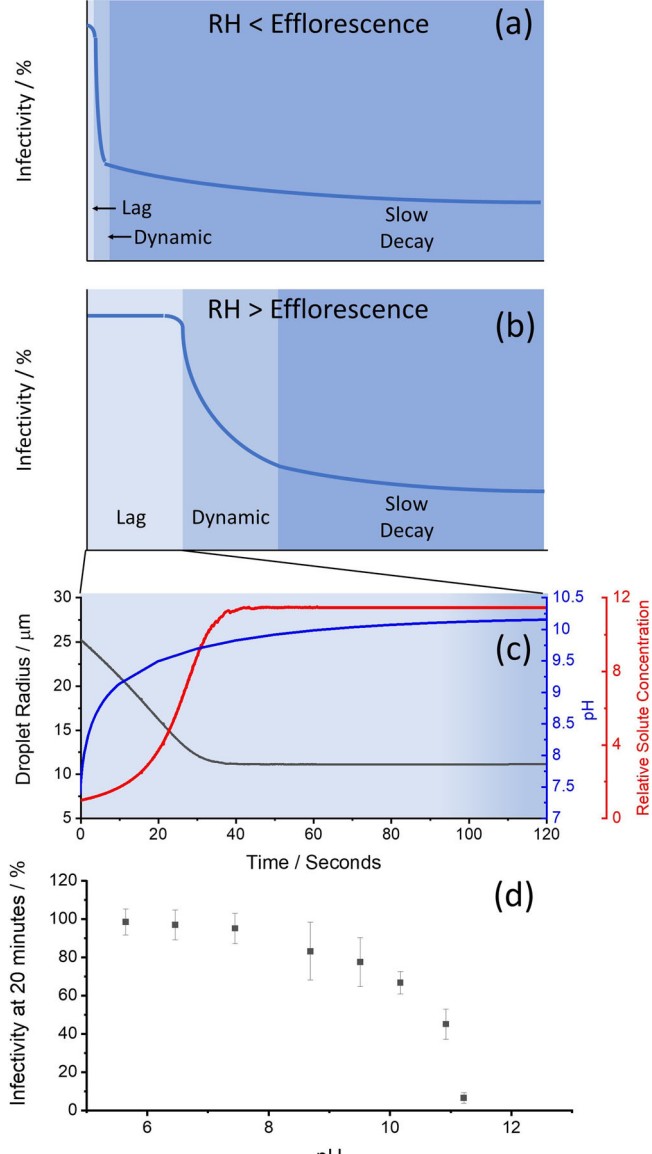

**Fig. 6 | Summary of the Triphasic Viral Aerosol Decay (TVAD) profile of respiratory aerosol.** The general regions of the viral decay are indicated in (**a**) and (**b**), and are governed by the conditions in the droplet (**c**). The three phases are the "Lag Phase", the "Dynamic Phase" and the "Slow Decay Phase". Conditions in the droplet (**c**) are estimates for a respiratory droplet injected into 90% RH at 500 ppm $[CO_{2(g)}]$ based upon previously published reports (14). The dynamics of the TVAD are governed by the virus's sensitivity to pH of the aerosol (**d**); data from previous study 14; values are means ± SE ($n = 24$ (independent samples)).

First is the "Lag Phase". During this phase, the conditions in the droplet are in flux (i.e., $CO_2$ evaporation, water evaporation, solute concentration), but the conditions are such that they are not yet toxic to the viral particle. The length of the Lag Phase is highly RH (Fig. 3a), temperature, $[CO_{2(g)}]$ (Fig. 3b), and variant dependent (Fig. 2b). At low RH, the Lag Phase is very short (<5 s) since droplet efflorescence will initiate rapid loss of viral infectivity (Fig. 6a). At high RH, the pH of the droplet increases during the Lag Phase, thus the time before the virus begins to decay is dependent on how sensitive the virus is to pH (Figs. 1a and 3a), and the speed at which the aerosol pH rises (Fig. 3b), a process that may take on the order of minutes. The wild type SARS-CoV-2 had a 2 min Lag Period. The virus was readily able to survive the rapid increase in salt concentration that took place over the first 20−30 s of aerosolization. Viral decay did not begin until the pH level

had considerably increased when it began to drive the loss of infectivity (Fig. 6d). It is likely that the increase in salt concentration plays a role in further increasing the rate of viral inactivation, but we suggest that the change in salt concentration alone is not the primary driving force based on the length of the Lag Period relative to the evaporation rate. The interplay between these two processes on viral stability needs to be explored further.

After the Lag Phase, the conditions in the droplet are still in flux, while the pH increase is still occurring. The second phase of decay is the "Dynamic Phase" phase. In this phase, the pH in the droplet has changed to the point where it has become toxic to the virus and the rate of loss is dictated by the changing conditions in the droplet. For example, the rapid loss of infectivity during efflorescence may occur in the Dynamic Phase. In the dynamic phase, the more susceptible fraction of the viral population will be inactivated.

The final phase is the "Slow Decay Phase" phase where the loss of viral infectivity is much slower than the Dynamic Phase. The Slow Decay Phase continues until all viral infectivity is lost, which may take tens of minutes to hours, depending on the virus and environmental conditions. The mechanism of the slower decay rate in this phase remains unclear and may be biological (e.g., fraction of strain being more resilient to the conditions) or physical/chemical (e.g., the remaining virus protected by its immediate location such as in a liposome or associated with a protein[26]). The length of the Lag Period and the rate of loss in the Dynamic Phase are both highly dependent on the pH sensitivity of the virus as well as $[CO_{2(g)}]$ and RH. At a high $[CO_{2(g)}]$, the Dynamic Phase can be dramatically truncated or even eliminated entirely (Fig. 2b). The decay rate in the Slow Decay Phase is largely RH independent, but highly dependent on the $[CO_2]$ (Figs. 2a, b, d and 3b). The data collected using closed systems, such as a Goldberg drum, largely miss the Lag and Dynamic phases (perhaps catching the tail end of the Dynamic Phase) and measure primarily in the Slow Decay Phase. Moreover, the rate of loss in the Slow Decay Phase is highly $[CO_{2(g)}]$ dependent (Figs. 2a, b, d, 3b and Supplementary Fig. 7).

The Triphasic Viral Aerosol Decay (TVAD) profile accurately describes the general relationship between viral infectivity and aerosolization time. The applicability of the TVAD to all other respiratory viruses is unconfirmed, though it is notable that we have observed similar behavior for the MHV virus and all VOC for SARS-CoV-2[14,26,37]. TVAD provides a framework to understand and explore which properties of the viral particle or the aerosol droplet will affect the likelihood of both short and long-distance transmission. The parameters that affect the Lag and Dynamic phases may affect short-distance transmission while the parameters that effect the Slow Decay Phase may affect long-distance transmission.

## Methods
### Cell culture and virus growth
Vero E6 cells modified to stably express TMPRSS2 (Vero E6/TMPRSS2 cell, obtained from NIBSC, UK) and Vero E6 cells modified to stably express human ACE2 and TMPRSS2 (Vero E6/ACE2/TMPRSS2 (VAT) cells (25)) are cultured in Dulbecco's Modified Eagle Medium (DMEM, high glucose; Sigma, UK) supplemented with 10% fetal bovine serum (FBS, Sigma) at 37 °C and 5% $CO_{2(g)}$, 100 units/ml penicillin (Gibco, UK), 100 μg/ml streptomycin (Gibco, UK), and L-glutamine (Gibco, UK).

Viral stocks of the B.1.617.2 (Delta) (GISAID ID: EPI_ISL_1731019) and Omicron (BA.2) VOCs are prepared using VAT cells grown to confluence in T75 flasks. The cells are infected at different multiplicities of infection (MOI) of 0.01 and 0.1 and the infected cells grown in Eagle's Minimal Essential Medium (MEM) supplemented with 2% FBS for either 24-h (for Delta) or 72-h (for BA.2) post-infection for virus production. After the incubation (37 °C, 5% $CO_2$) period, the culture supernatant is collected, centrifuged (250 g, 10 min), filtered (0.22 μm), aliquoted and frozen (−80 °C). The Delta VOC ($TCID_{50}$/mL of $3.4 \times 10^7$) grows to a higher titer than the BA.2 VOC ($TCID_{50}$/mL of $3.1 \times 10^6$).

## CELEBS – airborne longevity measurements

The airborne stability of SARS-CoV-2 variants is measured using the CELEBS technique. This has been described in detail previously for the study of both bacterial[23,24] and viral species[14,26,37]. Briefly, a starting formulation containing the virus is loaded into a droplet-on-demand dispenser. Depending on the magnitude and pulse width used to activate the dispenser, the starting radius of the droplet is ~25 microns. The number of infectious viral units per droplet is measured and found to be ~2.8 for the Delta VOC, ~0.3 for the Omicron BA.2, and ~0.5 for the Beta VOC. The final radius of the droplets will depend on RH, droplet composition, and initial size; with regards to the droplet compositions and sizes used in this study, our previous published work has shown the final radius between 5 and 10 $\mu$m in our previous work[14]. The dispenser is positioned near an induction electrode such that, during droplet generation, ion migration (e.g., $Na^+$) in the jet results in the production of an individual droplet with a slight net charge (<5fC from ion imbalance) when the jet collapses. The individual droplets are then levitated by the electrodynamic fields produced by the ring electrodes within the core of the CELEBS.

A laminar airflow is passed over the region in which the droplets are levitated such that the chemical composition, relative humidity, and temperature of the air can be controlled and monitored. The sources of compressed air used in the study were either $CO_2$-free compressed air (for experiments where the concentration of $CO_2$ was set to 0 ppm) or compressed laboratory air (Bambi Oil Free Compressor, Model VTS75D). The temperature and humidity of the airflow passing over the levitated droplets was measured using a Honeywell (HIH-4602-C Series) humidity sensor. The concentration of $CO_2$ in the airflow was measured using a GSS ExplorIR-M Low Power $CO_2$ Sensor (range 0 to 20,000 ppm, accurate within 70 ppm). After a set time period, ranging from <5 s through to 40 min, the electric fields are manipulated such that the droplets are deposited into a Petri dish filled with 6 mL of DMEM containing 2% FBS. Within 15 min post deposition (typically under 5 min), the 6 mL of virus-containing medium is added in 100 $\mu$L aliquots to the central 60 wells (containing confluent Vero E6/TMPRSS2 cells) of a 96-well plate. After 3–5 days incubation (in 37 °C, >99% RH, 5% $CO_2$), the number of wells showing the characteristic cytopathic effect (CPE) of SARS-CoV-2 are tabulated, and the number of viral particles that maintained their infectivity throughout the levitation process is calculated as previously described[37]. Each day, the infectivity (Eq. 4) is normalized to the infectious units per droplet of those levitated at 90% RH and <5 s (conditions under which no loss of viral infectivity is seen), termed the time (T) = 0 point.

$$\text{Infectivity} = \frac{\text{Virus per Droplet(Time,RH,[CO2])}}{\text{Virus per Droplet(<5s,90\%,500ppm)}} \quad (4)$$

The aerostability of different variants of SARS-CoV-2 are systematically measured across a broad range of environmental conditions (RH from 20% to 90%, $CO_2$ concentrations from 0 ppm to 6500 ppm) and time (from seconds to upwards of 40 min).

## Droplet solute selection

We previously reported that the physicochemical properties of aerosol particles composed of MEM and artificial saliva are similar (14, 15). As well as broad similarities in composition, these properties include the aerosol pH, which changes from neutral to >10 following generation (resulting from a similar initial bicarbonate concentration), as well as a variability in particle phase and morphology when the RH is between the deliquescence (~75%) and efflorescence (~50%) RHs of the aerosol. For MEM, all aerosol particles are homogeneous liquids containing no precipitate above an RH of 85%, while all particles contain a large proportion of crystalline salt below 45% RH. Indeed, initially formed droplets of MEM and saliva have similar concentrations of salts (to ensure a stable osmotic pressure for cell growth) and $NaHCO_3$

(because both are at equilibrium with ~5% $CO_2$). Thus, MEM can be considered a good proxy for saliva as they share the underlying physicochemical properties that govern viral aerostability. Indeed, we have shown that the infectivity recorded at high and low RH and over the first 120 s of levitation (enough time for the majority of the viral infectivity to be lost) there are not significant in their differences. As a consequence, MEM is used as the droplet medium in this study, and infectivity measurements were undertaken in conditions where the structure of the particles is known to be homogeneous, specifically at RHs of 40% and 90%.

## Bulk stability measurements

The stability of the Delta and BA.2 variants in bulk solutions is assessed either by $TCID_{50}$ assay detection of infectious virus or by immunostaining. In the $TCID_{50}$ assay, the virus stock is first diluted 1 in 10,000 in DMEM with 2% FBS which is adjusted to a pH of 11 with NaOH. After a set incubation period at 20 °C in the open air, the sample is then serially diluted into neutral DMEM containing 2% FBS, and the $TCID_{50}$/ml of each sample is then measured. For all samples, the $TCID_{50}$/ml is normalized to a sample that is not exposed to a change in the growth medium pH.

In the immunostaining approach, the virus stock is diluted 1:40 in DMEM with 2% FBS (pH 11). After a set incubation period, the sample is then diluted 1:15 in DMEM with 2% FBS (pH 7.0). 100 $\mu$L of the diluted virus is then added to the well of a 96-well plate seeded with Vero E6/TMPRSS2 cells the day prior. After an 18 h incubation (37 °C, 5% $CO_2$), the cells are fixed in 4% (v/v) paraformaldehyde for 60 min. Fixed cells are then permeabilized (0.1% Triton-X100 in PBS) and blocked (1% (w/v) bovine serum albumin) prior to being stained with a monoclonal antibody against the SARS-CoV-2 nucleocapsid protein (N) (1:2000 dilution; 200-401-A50, Rockland) followed by an appropriate Alexa Fluor-conjugated secondary antibody (ThermoFisher) and DAPI (Sigma Aldrich). Images of the wells are collected (ImageExpress Pico automated imaging platform[15]), and the number of infected cells tabulated.

## Statistical analysis

Error bars provided in the figures are reporting standard error. Significance was measured using a two-sample $t$-test with equal variance, where the $p$-values are reported as being less than 0.5, 0.05, or 0.005. When multiple variables were explored, a two-way ANOVA was used to assess the interaction between them (e.g., effect of $[CO_{2(g)}]$ and RH on aerostability).

## Reporting summary

Further information on research design is available in the Nature Portfolio Reporting Summary linked to this article.

# Data availability

The data generated in this study are provided in the Supplementary Information and the Source Data file. Data are available at the University of Bristol data repository, data.bris, at https://doi.org/10.5523/bris.17xvyth00473q2cxnj3ubg1vm7. Source data are provided with this paper.

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

## Acknowledgements

Vero E6 cells modified to stably express human ACE2 and TMPRSS2 were a kind gift from Dr Suzannah Rihn, MRC-University of Glasgow Centre for Virus Research, and cultured by Lisa Wright. Viral stocks of the B.1.617.2 (Delta) (GISAID ID: EPI_ISL_1731019) and Omicron (BA.2) VOCs were kindly provided by Professor Wendy Barclay, Imperial College, London, Professor Maria Zambon, UK Health Security Agency, and Dr Thushan de Silva, University of Sheffield. This work was funded by: the National Institute for Health Research-UK Research and Innovation (UKRI) rapid COVID-19 call, the Elizabeth Blackwell Institute for Health Research, the University of Bristol, and the Medical Research Council, the PROTECT COVID-19 National Core Study on Transmission and environment, managed by the Health and Safety Executive on behalf of Her Majesty's Government. The EPSRC Centre for Doctoral Training in Aerosol Science EP/S023593/1 which supported R.A. and J.T is acknowledged. A.E.H. and M.O.-F. received funding from the Biotechnology and Biological Sciences Research Council (BB/W00884X/1). J.F.R. is supported by funding from the Alexander von Humboldt Foundation. A.D.D. is a member of the G2P-UK National Virology consortium funded by the Medical Research Council/UKRI (Grant MR/W005611/1) that supplied SARS-CoV-2 variants. H.P.O. is supported by funding from the Defence Science and Technology Laboratory and the Engineering and Physical Sciences Research Council. S.J.C is supported by the Leverhulme Trust (Early Career Fellowship, ECF-2021-072) and the Isaac Newton Trust (20.08(r)). The authors would also like to acknowledge the critical role that the Bristol UNCOVER Group played in organizing this collaboration. The authors thank Sam Cobb (University of Cambridge) for their insights into $CO_2$ flux/aerosol pH.

## Author contributions

Author contributions: A.E.H., H.P.O., M.O.-F., J.F.S.M., A.F., A.D.D., T.C. and J.P.R. designed research; A.E.H., H.P.O., M.O.-F. and J.R. performed research; T.A.C. and A.D.D. contributed new reagents/analytic tools; H.P.O., A.E.H., M.O.-F., T.A.C., J.R. and J.P.R. analyzed data; A.E.H., H.P.O. and J.P.R. wrote the paper; A.E.H., H.P.O., J.F.S.M., D.J.H., A.F., A.D.D., J.R. and J.P.R. edited the paper.

## Competing interests

The authors declare no competing interests.
