## [Peer Review File · Nature Communications]

Ambient carbon dioxide concentration correlates with SARS-CoV-2 aerostability and infection riskEditorial Note: Parts of this Peer Review File have been redacted as indicated to remove third-party material where no permission to publish could be obtained.

REVIEWER COMMENTS

Reviewer #1 (Remarks to the Author):

The authors explore the effect of the concentration of CO₂ in the air on the survival of SARS-CoV-2 in aerosol particles. They accomplish this utilizing a system that they have published on previously that can trap and hold aerosol particles for extended periods of time. The trapped particles can be subjected to different conditions by varying the composition/conditions of air flowing past the particles.

The results demonstrate that the survival of the Delta variant of SARS-CoV-2 is increased at a particle age of 2 minutes at CO₂ levels of 2000 ppm or greater. The higher CO₂ levels are hypothesized to dampen an increase in particle pH post-generation that occurs with bicarbonate rich media due to a shift in the balance of the carbonic acid pathway, thereby preserving viral infectivity. The authors generalize this result to hypothesize that ambient CO₂ concentrations, which can vary significantly in indoor environments, may influence survival of SARS-CoV-2 and other respiratory viruses and, subsequently, the risk of infection in enclosed spaces.

While the results presented are interesting and potentially identify a novel pathway for viral inactivation in aerosol particles, there are numerous issues with the manuscript that need to be addressed prior to publication. These issues, as well as other comments, are detailed below:

General Comments:

- The authors do not uniformly provide the results of statistical analyses throughout the manuscript. P-values, number of replicates, and details of the statistical tests utilized are provided for some graphs/experiments, but not all, making interpretation of the data and associated graphs challenging at times. For example, the results state that the decay of the BA.2 variant was much slower at 90% RH than that observed for Delta. This statement is supported solely by saying that the amount of infectious virus remaining at an aerosol age of five minutes was 1.7 times higher for BA.2 versus Delta. No statistical analyses are provided to support this statement. Additional instances of a lack of statistical analysis are provided below. Additionally, in several instances where statistical analysis is provided, the tests chosen do not appear to be the most appropriate (i.e. using multiple t-tests in lieu of ANOVA).

- As noted above, the authors generalize the result observed with the Delta variant of SARS-CoV-2 to hypothesize that ambient CO₂ concentration is an important factor influencing the survival of SARS-CoV-2 and other respiratory viruses and, subsequently, the risk of infection in enclosed spaces. However, the authors also present data that suggest that there may be significant variability in pH sensitivity of different SARS-CoV-2 variants, with the more recent, and arguably more relevant, Omicron BA.2 subvariant of SARS-CoV-2 showing less sensitivity to pH changes in bulk suspension than the earlier Delta variant. Based on the authors' hypothesis and the decreased pH sensitivity of BA.2 subvariant, wouldn't it be expected that BA.2 in aerosol particles would be less sensitive to changes in atmospheric CO₂ than the Delta variant? If this were true and there was significant variability between SARS-CoV-2 variants in the sensitivity to pH changes/CO₂ levels, it seems inappropriate to extend this mechanism to other unrelated viruses, such as RSV or influenza.

Unfortunately, while differences in pH sensitivity between the Delta and Omicron variants were shown in bulk suspensions, the authors did not explore this comparison in aerosol particles, for reasons that are not completely clear (it is stated that the effect of CO₂ on the infectivity of aerosol particles was only explored with the Delta variant due to reasons related to viral titers and throughput, but this rationale is not completely clear, as they were able to make measurements with the BA.2 subvariant as part of the comparison shown in Figure 1A). Exploration of the effect of CO₂ concentrations on the survival of the Omicron variant in aerosol particles, and comparison of these results to those obtained with the Delta variant, are important as they would demonstrate the variability of the response to CO₂ across different variants and help to determine whether this mechanism can be generalized across variants or to other viruses, as the authors do. If a significant decrease in the pH sensitivity of the BA.2 variant relative to the Delta variant were

observed, it would suggest a potential mechanism that would help to explain the increased transmissibility associated with the Omicron variant relative to the Delta variant. Some additional discussion of these caveats and possibilities seem warranted.

- Many of the figures include substantial amounts of data. However, they are small and it is difficult to discern some of the text due to some of the shapes/colors utilized. Please consider revising to make the figures more easily readable.

Additional specific comments:

Introduction, Lines 31-32: It is stated that aerosols are the dominant route of transmission driving the spread of COVID. There is still debate about the relative contribution of various routes of exposure, and the reference cited does not provide strong evidence to back up this claim. Please consider softening this language.

Introduction, Lines 88-89: Please describe what the Controlled Electrodynamic Levitation and Extraction of Bioaerosol onto a Substrate is. I realize that the authors have published on this previously, but many readers of a cross disciplinary journal such as this are not likely familiar with this technology.

Methods, Lines 97-98: It is stated that all methods are included in the supplementary materials, presumably due to word limits. However, some description of the methods should be included here to aid the reader in understanding what was done instead of putting everything in supplemental materials. Placing methods completely in the supplementary materials seems at odds with the journal guidelines. In scanning multiple other articles in the journal, I did not see others that placed all methods in the supplementary material.

Results, Lines 101-116 and Figure 1A:

- How was the infectivity measured in Figure 1A?
- No statistics are included in this section to support the comparisons made. For example, it is stated that the amount of BA.2 present at 5 minutes is 1.7 times higher than Delta VOC, implying that BA.2 survives better, but no p-values or description of the analyses performed are present. Please update to include these, as well as the number of replicates for each point shown.
- Why are comparisons only made for a single time point when measurements were made across multiple time points? Would it not make more sense to utilize all of the data (perhaps excluding the data from the lag phase) to calculate a rate of loss (perhaps using linear regression on log transformed data), which could then be compared between the different permutations presented? A similar approach appears to be utilized later in the manuscript to model transmission.
- What is the rationale for plotting the standard error instead of standard deviation in this figure and others?

Results, Lines 126-138 and Figures 1B and 1C: The difference in pH sensitivity in the bulk phase is interesting. However, while the differences in stability are more apparent from the graphs for bulk phase measurements, statistics are again lacking for the comparisons presented here. A curve fit to the data to estimate the rate of loss seems like an appropriate way to compare the different VOCs.

Results, Lines 146-149: It is unfortunate that tests were not conducted with the BA.2 VOC. This would have been very informative given the dominance/fitness of the BA.2 VOC relative to the Delta VOC and would have nicely complemented the results related to pH sensitivity in bulk suspensions presented in Figure 1. As noted earlier, if the difference in pH sensitivity was also present in the aerosol phase, this difference could be a potential explanation for the increased viral transmissibility/fitness observed with BA.2.

For the particle sizes and viral titers utilized, what would be the expected viral load in a particle for each of the VOCs?

Results, Lines 151-152: This sentence is a bit confusing as written. Is it meant that the survival of virus after 2 minutes of aging/levitation is increased for the levels of CO₂ above 500 ppm? Please consider revising. Also, the next CO₂ level above 500ppm appears to be 2000ppm. Thus, the statement that "any increase in [CO₂(g)] results in a significant increase in viral aerostability" does not seem justified, as there are no data presented for intermediate levels.

Results, Lines 152-153: To clarify, the compressed, CO₂ free air corresponds to the 0 ppm point listed in Figure 2A? How was CO₂ concentration measured in the test system?
Is there a reason survival was assessed at 2 minutes for the different CO₂ levels, but 5 minutes in the previous comparison between the VOCs presented in figure 1?

Results, Lines 154-155: The lower p-value for the 6500 ppm dataset is likely the result of the decreased variability associated with this dataset relative to the 2000 and 3500 ppm datasets, and does not indicate that survival was increased at this level relative to the 2000 and 3500 ppm levels as implied by the text. The appropriate comparison here would be to compare the 2000, 2500, and 6500 ppm datasets by ANOVA with an appropriate post-test to determine if there were significant differences in the survival at 2 minutes between the different CO₂ levels.

Results, Figure 2A – As noted in the previous comment, ANOVA is likely more appropriate here given that multiple comparisons are being made. Additionally, how many replicates were performed at each CO₂ level, and what was the RH for these tests?

Results, Lines 167-169: How fast does CO₂ partition between the droplet and environment, and a steady state pH achieved in the particle? Is there a way to measure the pH of the particle during levitation? This statement says stability would be expected increase over time but does not give a time frame.

Results, Lines 169-173 and Figure 2B: No statistics are presented for the comparison between the different CO₂ levels that is discussed. Please include some statistical analysis to support the statement that CO₂ has a “considerable effect on the overall decay profile.” It appears that the data may fit an exponential decay profile, which would allow comparison of decay constants for the different CO₂ levels. The authors note later in the manuscript that the decay is likely multi-phasic, which may necessitate a multi-phase fit, but this appears to be the approach taken later in the manuscript when transmission was modeled.

Results, Line 179: “becoming more stable after 20 minutes” – what statistical metric is this statement based on?

Results, Lines 179-184 and Figure 2C: It isn’t completely clear what is being graphed here. Is this the ratio of the raw infectivity data from Fig 2B? Please provide more details on how this analysis was performed. Since there are multiple unpaired data points at each time for each CO₂ level, how were the ratios taken? What infectivity assay was utilized here? Again, the discussion of the data from Figure 2C does not include any statistical analysis to support the statements made in the text.

Results, Lines 186-190: The authors suggest that the analysis published by van Doremalen may be inappropriate since the decay profile of SARS-CoV-2 measured in the present study appears to be multi-phasic. However, as the authors note later in the manuscript, the methods used in the van Doremalen paper that the authors reference here likely represent only a portion of the decay profile, specifically the slow decay phase. Thus, for this situation, where only a single phase is being examined, expressing the stability as a single half-life seems appropriate, which was the approach taken in the van Doremalen reference. This discussion does not add much to manuscript, and it is recommended to remove this discussion on the van Doremalen analysis and focus on the results of the present study.

Results, Lines 203-204: The authors generalize the results with the Delta VOC and state that “CO₂ has a profound effect on the overall aerostability of SARS-CoV-2.” However, it needs to be pointed out that these data are for the Delta VOC, and, given the results presented in Figure 1, may not be applicable to other more recent variants, which may be less sensitive to changes pH. This is a major caveat to the results and precludes the generalization that CO₂ has a “profound” influence on all SARS-CoV-2 variants.

Results, Lines 209-211: Please include references for this sentence. Reference #1 may be appropriate here.

Results, Lines 217-226 and Figure 3A: The difference in RH sensitivity in the first 15 seconds

between the variants is interesting, and has implications for short range transmission. Were the other variants and MHV studied previously similar to BA.2 across all RH levels, or just below the efflorescence point? If the other variants (excluding Delta) and MHV were similar to BA.2 across all RH levels, this might suggest that the Delta variant is an outlier. Thus, results with Delta may not be generalizable across all SARS-CoV-2 variants.

Results, Line 233-234: "doubling of the remaining aerosolized viral load" – was this statistically significant?

Results, Lines 236-263 and Figure 3C: The data and discussion on the effect of salt concentration and humidity on the infectivity of the Delta variant seem tangential to the main focus of the paper. Recommend moving this to supplemental materials. This would potentially free up space to include details on the methods in the main text.

Results, Lines 272-274: The data from large particle aerosols were used to inform transmission risk for small particle aerosols. Is this appropriate given that Figure 3C demonstrates that particle size may be a factor influencing survival in aerosols?

Results, Lines 318-321: In order to estimate the probability of infection, wouldn't it be necessary to utilize a dose-response relationship? The infection probability calculation presented appears to follow the form of an exponential dose-response model, except that it appears to be missing a term for the probability of the organism causing an infection?

Results, Figure 4: In Figure 4C, after ~ 8 hours, the probability of transmission appears to increase by approximately 2% at the higher CO₂ level. In Figure 4D, this change appears to be less, with the change in the probability of transmission after 1000 minutes of only ~0.5%? These changes do not seem like a very large effect. Some additional discussion would be helpful here to put these changes in context for the reader.

Results, Line 353-354: "[CO₂(g)] is estimated to have a profound effect on overall risk". How much does risk increase? Please further discuss the rationale for this statement. There does not appear any quantitative analysis provided to support this statement.

Discussion, Lines 387-394: This seems like a bit of stretch given the data that are presented. No data are presented showing that a CO₂ concentration of 1000 ppm impacts survival of SARS-CoV-2 Delta. The lowest level evaluated above the 500-ppm baseline appears to be 2000 ppm.

Discussion, line 305 and line 426: Please define "moderate increases" and "small increases". The data only show an increase in the stability of SARS-CoV-2 Delta in aerosol particles at CO₂ levels of 2000 ppm and above.

Discussion, line 417-419: This is an interesting observation, and it would be interesting to measure how much CO₂ is produced during nebulization of viral media as is done in some of the referenced studies.

Discussion, line 432: The summary of the triphasic decay profile is useful and a good summary of the process that occurs during particle equilibration.

Discussion, line 484-485: "Moreover, the rate of loss in the Slow Decay Phase is highly [CO₂(g)] dependent". In examining Figure 2B, if the rate of the slow decay is estimated as the slope of the line between the 20 and 40 minute time points, the slopes do not appear to differ greatly between the 500 and 3000 ppm CO₂ levels, which would seem to contradict this statement. However, again, a lack of statistical analysis precludes ready interpretation and evaluation of this statement.

Reviewer #2 (Remarks to the Author):

This manuscript provides new data on changes to the aerostability of SARS-CoV-2 aerosol particles due to acidification from dissolution of CO₂. This data represents a significant advancement in our understanding of the transmission of SARS-CoV-2 and potentially many other respiratory viruses. Overall, I find the work to be of high quality and well written. I have a few minor issues that I would like to see addressed prior to publication.

Issues

1. There should be more of a caveat on the issue of particle size. When performing the Well's Riley models, the authors make the assumption that the decay observed for the relatively large aerosol particles (17-25 micrometers) in this study would be extensible to smaller particles, such as those that might be produced in the bronchi or larynx. I think this is overstated. There will be size-dependent impacts due the different surface to volume ratios of the smaller particles that may change the rates fairly significantly. While I think the assumption for the purpose of making a point with the Wells Riley model is acceptable, there should be some additional attention paid to the issue of size-dependence.
2. Also, on the topic of aerosol size, the authors mention the initial size of the droplet, but don't report the expected resulting size range after evaporation, which is likely RH dependent. I graph of mean sizes, as a function of RH might be helpful.
3. The authors make the comments that "A survey of the literature found that a large portion of manuscripts do not provide the nebulization period, while none have been found to report the [CO₂(g)]." No articles are cited here, and no indication of how the survey was conducted is included. It might be easier to simply highlight the absence of this data from some key papers, that should be cited, and leave it at that.

**Ambient Carbon Dioxide Concentration Correlates
with SARS-CoV-2 Aerostability and Infection Risk**

by Haddrell et al.

Review by Thomas Peter

This is an interesting manuscript on the physicochemical properties of respiratory aerosol, providing evidence that CO₂-induced changes in particle pH can influence the infectivity of airborne SARS-CoV-2 viruses. The same group has previously shown that particles that mimic expiratory aerosol, albeit with much larger initial size (~50 µm), lose a significant fraction of dissolved bicarbonate as CO₂ upon exhalation, allowing their pH to rise from neutral to alkaline pH 11. To the degree airborne viruses are sensitive to alkaline conditions, this physicochemical change, which occurs within a few minutes after the drops are exposed to the gas phase, may lead to their inactivation. In this new manuscript, the authors generalize this finding by asking whether differences between freshly ventilated rooms (with a few hundred ppm of CO₂) and badly ventilated rooms with stale air (with a few thousand ppm) may influence virus inactivation. They could therefore be an air hygiene argument for good ventilation. They further speculate about how climate change with increasing CO₂ levels could affect the activity of such viruses in the future. The experimental work is based on their state-of-the-art electrodynamic balance apparatus (CELEBS) and uniquely applied to the aerostability of the SARS-CoV-2 Delta and Omicron variants.

Major concerns

While this is all great, I am concerned that this manuscript neglects key limitations of their study: (i) the exclusive focus on the effects of CO₂ in laboratory air, excluding minor trace gases that normally present in indoor air and more relevant for viral activity and transmission; (ii) exclusive use of particles larger than 20 µm (diameter in equilibrium with ambient air) while particles of less than 1 micron to a few micrometers are most relevant for infection and spread of viral respiratory infections; (iii) the use of minimal essential media (MEM or DMEM) as matrix for the experiments, which has a significantly higher salt content than, for example, nasal mucus.

In combination, these three aspects may make the inactivation of SARS-CoV-2 very different from what is described in this manuscript, and these caveats are not at all addressed in the manuscript.

I continue to think that this will be an interesting paper, but it requires discussing these constraints and correctly acknowledging other literature, and probably belongs into a more specialized journal than Nature Communications. Furthermore, the news value is somewhat lowered because the basic information about CO₂ evaporating from the droplets in their experiments leading to alkalization was recently published (Oswin et al., PNAS 2022).

Specific comments

Below I address the points above in more detail combined with other points line by line.

L. 18-20. Abstract. “Previous studies have shown that a rapid increase in the pH of respiratory aerosols following generation due to changes in the gas-particle partitioning of pH buffering bicarbonate ions and carbon dioxide is a significant factor reducing viral infectivity.” No, previously this has been shown (Oswin et al., PNAS, 2022) for MEM/DMEM particles of sizes larger than respiratory particles and under conditions without the trace gases in typical indoor air (e.g., HNO₃, NH₃, HCl, CH₃COOH). Therefore, I think this statement in the abstract is misleading. As Luo et al. (ES&T 2022) show, particles with the characteristics of nasal mucus and sizes more representative of exhaled aerosol particles emitted into typical indoor air turn acidic after typically a minute.

L. 31-33. Intro. “The inhalation of respiratory aerosol containing ... SARS-CoV-2 has been identified as the dominant route of transmission driving the spread of ... COVID-19 (Duval et al., BMJ, 2022)”. In fact, Duval et al. do not mention the “dominant route”. Rather, they conclude very carefully: “This rapid systematic review found evidence suggesting that long distance airborne transmission of SARS-CoV-2 might occur in indoor settings such as restaurants, workplaces, and venues for choirs, and identified factors such as insufficient air replacement that probably contributed to transmission. These results strengthen the need for mitigation measures in indoor settings, particularly the use of adequate ventilation.”

L. 45-46. This sentence states that singing and talking *reduce* the total number of virus-containing aerosol droplets. I believe the authors meant to express the opposite.

L. 68-70. The authors write: “What is clear is that models (Luo et al., ES&T 2022) and measurements (19, 20) of human exhaled aerosol have both shown consistently that exhaled respiratory aerosol is significantly more alkaline than the fluids within the respiratory tract from which they originate” (where 19, 20 refer to condensed breath condensate measurements). This statement is a distortion of the facts. Rather, Luo et al. show with biophysical modeling for synthetic lung fluid and nasal mucus that exhaled particles of initial radius of 1 μm start out slightly acidic (pH 6.6) in the respiratory tract, undergo very slight alkalization within very the first 0.2 s after exhalation reaching neutral pH (this is the CO₂ effect discussed by the authors), and then acidify to pH 6 after 1 s, pH 5 after 10 s, and finally pH 4 after 100 s. See Figs. 3F and 4A of the work by Luo et al. Their findings are therefore quite the opposite of “significantly more alkaline”. I suggest that the authors double-check the veracity of the claims made based on the existing literature

L. 68-70. The sentence “what is clear is that...” contains a further misunderstanding by assuming that measurements of breath condensate would prove that the aerosol is becoming alkaline. This is not the case. Doctors typically bubble the freshly collected breath condensate with argon for 10 minutes to remove the high amounts of dissolved CO₂ and bicarbonate. Hence, it is not surprising that the pH values reported for breath condensates are alkaline, but this has nothing to do with the evolution of pH of particles exhaled into indoor air.

L. 79. “Infectivity decay profile” – what is this?

L. 119. Figures: Please show at least one example with the individual measurements of the biological replicas instead of mean and standard errors, as only then one can properly judge the uncertainties. How is it possible that uncertainties are so small when determined

reductions are only a few 10 %? In many other investigations, including our own, when reductions are 2 orders of magnitude, error bars are much larger.

Figure 1. I found it hard to understand the caption. The first panel is introduced by (A) and then the explanation, the other panels first with the explanation, then by (B) and (C). Please indicate also the size of particles, this information is hard to find. I figure that (B) and (C) are both for RH 90 %? If so, why do infectivities not continue to decrease after 20 minutes?

L. 134. The statement that “collectively, the data shown in Figure 1 support the hypothesis that high pH achieved in aerosol” would be more convincing if the figure contained also some results for lower pH.

Figure 2. The information that the entire figure refers to RH 90 % is hidden in the ordinate of panel (A).

Figure 2B. Why is the decay of infectivity between 20 and 40 minutes faster for 500 ppm than for 3000 ppm? Can this be fully explained by the increase in aerostability through CO₂?

Figure 2D. Something is wrong with the ordinate $\ln([A]/[A_0])$. Why do the measurements start at $\exp(4.5)$ then approach a value of 1?

L. 167-169. “With the rate of loss of viral infectivity increased by elevated aerosol alkalinity, any improvement in aerostability of SARS-CoV-2 resulting from elevated [CO₂(g)] would be expected to increase over time with a greater tendency towards neutral pH due to the dissolution of carbonic acid.” I read this several times but cannot understand.

Figure 3. I did not grasp how the dependence on NaCl concentration is related to the main topic of the manuscript.

L. 254. A “solute salt crystal” – what is this?

L. 258-261. Entangled style of writing, hard to decipher.

L. 270. Please define “gas phase like”.

L. 405-406. “Moderate increases in [CO₂(g)] affecting the aerostability of SARS-CoV-2 have very broad implications with regards to how all previously published aerovirology experiments should be interpreted.” While I fully agree that the community needs to pay more attention to the composition of the gas phase beyond temperature and relative humidity, I find this an overstatement and think attention also needs to be directed to other air components, such as HNO₃, NH₃, HCl, CH₃COOH, which besides the carbonate buffer introduce chloride buffer, ammonia buffer, etc.

L. 585. Something is wrong with the citation.

In the following, the Reviewer's comments are identified in *italics*, our response in **bold**, and changes made to the manuscript are in normal text and indented; the location of the updated text are identified by the section and paragraph in which they were made.

Reviewer 1

The authors explore the effect of the concentration of CO₂ in the air on the survival of SARS-CoV-2 in aerosol particles. They accomplish this utilizing a system that they have published on previously that can trap and hold aerosol particles for extended periods of time. The trapped particles can be subjected to different conditions by varying the composition/conditions of air flowing past the particles.

The results demonstrate that the survival of the Delta variant of SARS-CoV-2 is increased at a particle age of 2 minutes at CO₂ levels of 2000 ppm or greater. The higher CO₂ levels are hypothesized to dampen an increase in particle pH post-generation that occurs with bicarbonate rich media due to a shift in the balance of the carbonic acid pathway, thereby preserving viral infectivity. The authors generalize this result to hypothesize that ambient CO₂ concentrations, which can vary significantly in indoor environments, may influence survival of SARS-CoV-2 and other respiratory viruses and, subsequently, the risk of infection in enclosed spaces.

While the results presented are interesting and potentially identify a novel pathway for viral inactivation in aerosol particles, there are numerous issues with the manuscript that need to be addressed prior to publication. These issues, as well as other comments, are detailed below:

General Comments:

- The authors do not uniformly provide the results of statistical analyses throughout the manuscript. P-values, number of replicates, and details of the statistical tests utilized are provided for some graphs/experiments, but not all, making interpretation of the data and associated graphs challenging at times. For example, the results state that the decay of the BA.2 variant was much slower at 90% RH than that observed for Delta. This statement is supported solely by saying that the amount of infectious virus remaining at an aerosol age of five minutes was 1.7 times higher for BA.2 versus Delta. No statistical analyses are provided to support this statement. Additional instances of a lack of statistical analysis are provided below. Additionally, in several instances where statistical analysis is provided, the tests chosen do not appear to be the most appropriate (i.e. using multiple t-tests in lieu of ANOVA).

A significant portion of Reviewer #1 comments are centred on our use of statistics throughout the manuscript. In response, we have sought advice from a statistician to assist our data analysis and re-evaluated the way in which many of the differences have been reported. None of the conclusions drawn in the original submission (i.e. assertions that trends are, or are not, significant) have changed from our re-evaluation. Rather, the methods used to assert significance have been improved through use of the methods suggested by Reviewer #1. Their suggestion to improve our statistical analysis is appreciated.

For the specific claim that the amount of viable BA.2 was 1.7 times higher than that of the Delta variant after 5 minutes, we have added statistical evidence that this is a significant effect (details further below). Rather than summarizing the other changes made in the manuscript here, we will address the specific concerns as they are raised in our responses below.

- As noted above, the authors generalize the result observed with the Delta variant of SARS-CoV-2 to hypothesize that ambient CO₂ concentration is an important factor influencing the survival of SARS-CoV-2 and other respiratory viruses and, subsequently, the risk of infection in enclosed spaces. However, the authors also present data that suggest that there may be significant variability in pH sensitivity of different SARS-CoV-2 variants, with the more recent, and arguably more relevant, Omicron BA.2 subvariant of SARS-CoV-2 showing

less sensitivity to pH changes in bulk suspension than the earlier Delta variant. Based on the authors' hypothesis and the decreased pH sensitivity of BA.2 subvariant, wouldn't it be expected that BA.2 in aerosol particles would be less sensitive to changes in atmospheric CO₂ than the Delta variant? If this were true and there was significant variability between SARS-CoV-2 variants in the sensitivity to pH changes/CO₂ levels, it seems inappropriate to extend this mechanism to other unrelated viruses, such as RSV or influenza.

Unfortunately, while differences in pH sensitivity between the Delta and Omicron variants were shown in bulk suspensions, the authors did not explore this comparison in aerosol particles, for reasons that are not completely clear (it is stated that the effect of CO₂ on the infectivity of aerosol particles was only explored with the Delta variant due to reasons related to viral titers and throughput, but this rationale is not completely clear, as they were able to make measurements with the BA.2 subvariant as part of the comparison shown in Figure 1A). Exploration of the effect of CO₂ concentrations on the survival of the Omicron variant in aerosol particles, and comparison of these results to those obtained with the Delta variant, are important as they would demonstrate the variability of the response to CO₂ across different variants and help to determine whether this mechanism can be generalized across variants or to other viruses, as the authors do. If a significant decrease in the pH sensitivity of the BA.2 variant relative to the Delta variant were observed, it would suggest a potential mechanism that would help to explain the increased transmissibility associated with the Omicron variant relative to the Delta variant. Some additional discussion of these caveats and possibilities seem warranted.

The reviewer is correct that the sensitivity of the viral variant to high pH will dictate the degree to which CO₂ buffering of the aerosol will affect aerostability. This means that the Delta variant of concern (VOC) will never be more aerostable than the BA.2 VOC, but the Delta VOC may decay at a similar rate under certain conditions (high enough [CO₂]). In the original submission, the dependence on CO₂ was only measured for the Delta variant due to the lower titres of the Beta and Omicron variants that could be achieved in the viral stock solution. Given the concerns raised by the reviewer, we have made a renewed effort to overcome this limitation. Due to these efforts, the dependencies of aerosol stability of the Omicron BA.2 and Beta VOCs on increased [CO₂] have now been quantified (revised Figure 2A). As expected, increasing the [CO₂] significantly increased the aerostability of both VOCs. Figure 2A has been replaced with the revised figure below:

Figure 2 Exploring the effect of [CO₂(g)] on the aerostability of the Delta VOC as measured with the CELEBS. (A) Infectivity of the Delta, Beta and Omicron BA.2 VOCs as a function of ambient

concentrations of CO₂ at 90% RH, 120s. Statistical significance was assessed using a two-sample equal-variance t-test (*p ≤ 0.05, **p ≤ 0.005).

The following discussion of the implications and caveats of these new data has been added below Figure 2 (Results: SARS-CoV-2 Aerostability Correlates with the Ambient Concentration of Gas Phase Carbon Dioxide, 3rd paragraph):

Increasing the [CO_{2(g)}] resulted in an increase in the aerostability of both the Beta and Omicron BA.2 VOCs. This demonstrates that the viral aerostability is dependent on CO_{2(g)} concentration for all SARS-CoV-2 variants. At 120s, the increase in viral infectivity due to the increase of [CO_{2(g)}] was similar across all variants. Given the differing pH sensitivities for different variants, it is unlikely a similar increase across variants may not occur at all time periods for all variants.

Regarding extending the mechanism to other viruses, there are simply not many studies that explore the sensitivity of viruses in high pH (>10). We feel that the open question is not whether this mechanism can be extended to other viruses, but to what degree other viruses are affected by this mechanism and if there are additional mechanisms yet to be identified that play a role. We have currently secured funding to further explore this question.

- Many of the figures include substantial amounts of data. However, they are small and it is difficult to discern some of the text due to some of the shapes/colors utilized. Please consider revising to make the figures more easily readable.

The text size in figures 4 and 5 has been increased.

Additional specific comments:

Introduction, Lines 31-32: It is stated that aerosols are the dominant route of transmission driving the spread of COVID. There is still debate about the relative contribution of various routes of exposure, and the reference cited does not provide strong evidence to back up this claim. Please consider softening this language.

Both Reviewers #1 and #3 raised this concern. As recommended, the language was moderated and now reads (Introduction, 1st paragraph):

The inhalation of respiratory aerosol containing the severe acute respiratory syndrome coronavirus-2 (SARS-CoV-2) has been identified as an important route of transmission in the spread of coronavirus disease 2019 (COVID-19) (1).

Introduction, Lines 88-89: Please describe what the Controlled Electrodynamic Levitation and Extraction of Bioaerosol onto a Substrate is. I realize that the authors have published on this previously, but many readers of a cross disciplinary journal such as this are not likely familiar with this technology.

The brief description of the CELEBS has been moved from the Supplemental Information to the Methods section. Accordingly the Methods section has been significantly expanded. Moreover, a brief overview to the technique has been added to the introduction (Introduction, 6th paragraph):

The CELEBS is a next generation aerobiology technique that allows for changes in the viral infectivity within a small population of levitated droplets of a near identical size and composition to be measured as a function of time, temperature, relative humidity and complete gas phase

composition. Additionally, viral decay from <5s to hours can be readily measured with this technique.

Methods, Lines 97-98: It is stated that all methods are included in the supplementary materials, presumably due to word limits. However, some description of the methods should be included here to aid the reader in understanding what was done instead of putting everything in supplemental materials. Placing methods completely in the supplementary materials seems at odds with the journal guidelines. In scanning multiple other articles in the journal, I did not see others that placed all methods in the supplementary material.

As with the above comment, a brief summary of both the CELEBS and the experimental techniques used have been added to the Methods section. A more complete description of the methods remains in the Supplementary Information section.

Results, Lines 101-116 and Figure 1A:

- *How was the infectivity measured in Figure 1A?*

This was originally described in the Supplementary Information. As with the above two comments, a brief summary has now been included in the Methods section of the manuscript. Discussion of how infectivity is defined has also been added. An equation showing how infectivity is reported from CELEBS measurements has also been added to the Methods section.

- *No statistics are included in this section to support the comparisons made. For example, it is stated that the amount of BA.2 present at 5 minutes is 1.7 times higher than Delta VOC, implying that BA.2 survives better, but no p-values or description of the analyses performed are present. Please update to include these, as well as the number of replicates for each point shown.*

We have updated the manuscript to address this point and to state the number of degrees of freedom for each point in the table in the Supplemental Information section (Supplemental Table 1). The relevant passage now reads (Results: The BA.2 Omicron VOC is More Aero-Stable than the Delta VOC, 1st paragraph):

Excluding data points in the transient decay (< 1 minute) to focus on the more gradual decay from 2 minutes onwards, our analysis revealed significant differences in infectivity between the Delta and BA.2 VOCs. At 2 minutes, we observed a statistically significant difference in the infectivity of the two variants of $26 \pm 6\%$ ($p=3 \times 10^{-4}$, with 19.8 effective degrees of freedom via Satterthwaite approximation). This effect persisted at the 5-minute mark, with a difference of $19 \pm 8\%$ ($p=0.03$, with 18.9 degrees of freedom).

- *Why are comparisons only made for a single time point when measurements were made across multiple time points? Would it not make more sense to utilize all of the data (perhaps excluding the data from the lag phase) to calculate a rate of loss (perhaps using linear regression on log transformed data), which could then be compared between the different permutations presented? A similar approach appears to be utilized later in the manuscript to model transmission.*

The reviewer is correct that comparisons between the time-dependent trends would be ideally compared across the multiple decay profiles presented in this study (Figures 1A, 2B). However, this is not a robust approach as it does not account for the complex viral decay dynamics (that are not single exponential), with different mechanisms coming to the fore at different RHs. As shown in Figure 2D, the decay profiles of

the virus in the air do not follow a simple first order decay, so the slopes of those curves cannot be simply compared. To exclude the initial lag phase as the reviewer suggests would exclude a large, and no doubt important, fraction of the decay profile. Thus, the decision was made to report the significance at each time point.

- *What is the rationale for plotting the standard error instead of standard deviation in this figure and others?*

Our focus is on accurately reporting the general trend in viral infectivity over time and are less concerned with reporting the noise within the data itself. To provide the reader with further context, the number of samples per data point have been added to the figures (Supplemental Table 1). Additionally, using standard error is common in the area of aerovirology, thus the error bars were left as standard error to make our study easily comparable to others published in field.

Results, Lines 126-138 and Figures 1B and 1C: The difference in pH sensitivity in the bulk phase is interesting. However, while the differences in stability are more apparent from the graphs for bulk phase measurements, statistics are again lacking for the comparisons presented here. A curve fit to the data to estimate the rate of loss seems like an appropriate way to compare the different VOCs.

The infectivity profiles for the bulk phase measurements in Figure 1B and 1C show first order decay kinetics reflecting that the decrease in infectivity in aerosol measurements results from a much more complex interplay of processes. We have converted the data in Figures 1B and 1C to $\ln(\text{Infectivity})$ vs time, and the significance of difference for the linear fits has been calculated. For both 1B and 1C, the difference in slope between Omicron and Delta is significant ($p=0.0002$ for (B) and $p=0.027$ for (C)).

The following has been added to the Figure 1 caption:

Both (B) and (C) fit a first order decay; the linear fits of $\ln(\text{infectivity})$ vs. time for the Omicron and Delta VOCs are consistent with decay rates that are significant in their difference ($p=0.0002$ for (B), $p = 0.027$ for (C)).

Results, Lines 146-149: It is unfortunate that tests were not conducted with the BA.2 VOC. This would have been very informative given the dominance/fitness of the BA.2 VOC relative to the Delta VOC and would have nicely complemented the results related to pH sensitivity in bulk suspensions presented in Figure 1. As noted earlier, if the difference in pH sensitivity was also present in the aerosol phase, this difference could be a potential explanation for the increased viral transmissibility/fitness observed with BA.2.

We now report additional measurements on the BA.2 and Beta VOCs. As shown in Figure 1A, the Omicron BA.2 variant is more aerostable than the Delta variant at the same concentration of CO_2 . This is now discussed in line 112 of the original manuscript. Indeed, all variants are more resistant to inactivation at a higher CO_2 concentration (3,000 ppm) than the Delta variant. This is now shown in data added to Figure 2A.

For the particle sizes and viral titers utilized, what would be the expected viral load in a particle for each of the VOCs?

We only quantified the number of infectious viral particles per droplet as those are the ones associated with disease transmission. We have added the number of infectious virus particles per droplet as a function of each variant into the Methods section (Methods, 2nd paragraph):

The number of infectious viral units per droplet were measured, and found to be ~2.8 for the Delta VOC, ~0.3 for the Omicron BA.2, and ~0.5 for the Beta VOC.

Results, Lines 151-152: This sentence is a bit confusing as written. Is it meant that the survival of virus after 2 minutes of aging/levitation is increased for the levels of CO₂ above 500 ppm? Please consider revising. Also, the next CO₂ level above 500ppm appears to be 2000ppm. Thus, the statement that “any increase in [CO₂(g)] results in a significant increase in viral aerostability” does not seem justified, as there are no data presented for intermediate levels.

We have now performed and included additional measurements to help identify the increase in CO₂ concentration that leads to a detectable difference in viral aerostability. The new data are shown in Figure 2A. We found that increasing the [CO₂] from 500 ppm to just 800 ppm results in a significant increase in virus stability for the Delta VOC. Evidence that such a modest increase in [CO₂] can increase aerostability increases the importance of this study; although a concentration of 2000 ppm is only commonly found in poorly ventilated spaces, 800 ppm is a value much more commonly experienced indoors and, indeed, is often used as the threshold below which air quality is described as “good”. Furthermore, the impact of such a small increase of CO₂ provides evidence that even ambient CO₂ levels expected over the next century can result in increases in viral aerostability making it harder to prevent indoor carbon dioxide concentrations reaching a level that could increase disease transmission rates, even with good ventilation. This result also provides insights into why data collected using rotating drums report such long viral decay rates. The manuscript has been changed to highlight this critical finding, and now reads (Results: SARS-CoV-2 Aerostability Correlates with the Ambient Concentration of Gas Phase Carbon Dioxide, 2nd paragraph):

When compared to a typical atmospheric [CO₂(g)] (~500 ppm), increasing the [CO₂(g)] to just 800 ppm resulted in a significant increase in viral aerostability after 2 minutes (Delta VOC, Figure 2A). It is notable that, according to the UK Scientific Advisory Group for Emergencies (SAGE), 800 ppm [CO₂(g)] has been identified as the level below which a room is determined to be well ventilated. When ambient air flow into the CELEBS was substituted with synthetic air (0 ppm [CO₂]), no change in virus aerostability was observed.

Results, Lines 152-153: To clarify, the compressed, CO₂ free air corresponds to the 0 ppm point listed in Figure 2A? How was CO₂ concentration measured in the test system?

A cylinder of CO₂-free compressed air was used to collect the 0ppm point in Figure 2A. The following paragraph has been added to the Methods section in the Supplementary Information (Materials and Methods: CELEBS – Airborne Longevity Measurements, 1st paragraph):

The sources of compressed air used in the study were either CO₂-free compressed air (for experiments where the concentration of CO₂ was set to 0 ppm) or compressed laboratory air. The temperature and humidity of the airflow passing over the levitated droplets was measured using a Honeywell (HIH-4602-C Series) humidity sensor. The concentration of CO₂ in the air flow was measured using a GSS ExplorIR-M Low Power CO₂ Sensor (range 0 to 20,000 ppm, accurate within 70 ppm).

Is there a reason survival was assessed at 2 minutes for the different CO₂ levels, but 5 minutes in the previous comparison between the VOCs presented in figure 1?

In Figure 1A, the differences between the overall decay dynamics of the 2 variants was shown over 5 minutes (not just at a single time point). In order to measure the effect of a single variable (such as [CO₂])

on aerostability, we set the time point of interest at 2 minutes and fixed the others parameters. Ideally, the timepoint selected should be one for which either an increase or decrease in viability can be readily observed in response to the parameter. For the Delta variant, ~70% of the viral decay occurs in the first 2 minutes regardless of relative humidity; after that time period, the decay rate slows. Thus, it makes sense to explore the effect of CO₂ at a time point when much of the viral infectivity is lost.

Results, Lines 154-155: The lower p-value for the 6500 ppm dataset is likely the result of the decreased variability associated with this dataset relative to the 2000 and 3500 ppm datasets, and does not indicate that survival was increased at this level relative to the 2000 and 3500 ppm levels as implied by the text. The appropriate comparison here would be to compare the 2000, 2500, and 6500 ppm datasets by ANOVA with an appropriate post-test to determine if there were significant differences in the survival at 2 minutes between the different CO₂ levels.

The reviewer is correct in that the lower p-value is due to the reduced variability at higher [CO₂], though on re-analysis this appears to be because of a larger sample size at higher concentrations. We re-analysed these datasets with ANOVA which suggests that the datasets for 1800 ppm and above were not significantly different (p=0.53). Thus, we have clarified this statement in the revised manuscript (Results: SARS-CoV-2 Aerostability Correlates with the Ambient Concentration of Gas Phase Carbon Dioxide, 2nd paragraph):

When compared to a typical atmospheric [CO_{2(g)}] (~500 ppm), increasing the [CO_{2(g)}] to just 800 ppm resulted in a significant increase in viral aerostability after 2 minutes (Delta VOC, Figure 2A). No significant difference in infectivity was reported between 800 ppm and 6,500ppm.

To further address the concerns raised here by the reviewer, we have deleted the sentence: “It is notable that whilst the increase in [CO_{2(g)}] to ~6,500 ppm resulted in a more significant increase in aerostability, it did not result in complete stabilization of virus in the aerosol phase (Figure 2A).”

Results, Figure 2A – As noted in the previous comment, ANOVA is likely more appropriate here given that multiple comparisons are being made. Additionally, how many replicates were performed at each CO₂ level, and what was the RH for these tests?

The number of replicates for each data point has been added to a table (Supplemental Table 1) in the results section. The RH was 90% for these measurements, as indicated on the axis of the figure. We have also updated the figure caption to state the RH.

Results, Lines 167-169: How fast does CO₂ partition between the droplet and environment, and a steady state pH achieved in the particle? Is there a way to measure the pH of the particle during levitation? This statement says stability would be expected increase over time but does not give a time frame.

The rate of pH change in aerosol is difficult to ascertain. This is made even more challenging due to the complexities of composition within respiratory aerosol. The ability to accurately measure the pH change as a result of bicarbonate/CO₂ evaporation in a respiratory droplet is impossible at this time. However, based on the amount of bicarbonate in the starting formulation, one can estimate the maximum the droplet pH will reach (between 10.5 and 11).

Gas flux from an aerosol droplet is considered to be a very rapid process (<1s). However, bicarbonate evaporation via H₂CO₃ production has an additional hydration step, meaning that bicarbonate is not rapidly interconverted with CO₂ due to its requirement to (de)hydrate H₂CO₃ to form CO₂ gas (<https://www.nature.com/articles/s41557-021-00880-2>). This is not a mass transport restriction, it is a

chemical kinetics restriction (rate= approx 0.05 s⁻¹) as opposed to a diffusion limited reaction. These kinetics result in the time response being much slower than based simply on gas-particle equilibration rates. Collectively, this means that the time taken for the droplet to reach a maximum pH is on the order of minutes, not seconds. The precise time this process takes is unknown, and the subject of debate.

Lines 167-169 were slightly altered and the above citation added. The lines now read (Results: SARS-CoV-2 Aerostability Correlates with the Ambient Concentration of Gas Phase Carbon Dioxide, 4th paragraph):

The rate of viral infectivity loss correlates with aerosol alkalinity. Elevated [CO₂(g)] limits the amount of bicarbonate leaving the droplet (Eq. 1), thus limits the maximum pH that the droplet will reach. A significant improvement in aerostability of SARS-CoV-2 resulting from elevated [CO₂(g)] would be expected to increase over time as the droplet will spend less time at the elevated pH (28).

Results, Lines 169-173 and Figure 2B: No statistics are presented for the comparison between the different CO₂ levels that is discussed. Please include some statistical analysis to support the statement that CO₂ has a “considerable effect on the overall decay profile.” It appears that the data may fit an exponential decay profile, which would allow comparison of decay constants for the different CO₂ levels. The authors note later in the manuscript that the decay is likely multi-phasic, which may necessitate a multi-phase fit, but this appears to be the approach taken later in the manuscript when transmission was modelled.

As discussed, the aerosol data do not fit a single exponential decay profile. In Figure 2D, an exponential function would be a straight line on log-linear axes and it is clear that this is not the case. Thus, in order to establish that the decay profiles are different, a t-test was used to establish the significance of the difference between the total infectivity at high and low [CO₂] at each time point for the Delta VOC. It was found that for every time point from 2 minutes onward, the difference is significant (p<0.05). The following statement has been added to the Figure 2 caption:

Elevating the [CO₂] results in a significant difference in overall decay (p < 0.05) of the Delta VOC from 2 minutes onward.

Results, Line 179: “becoming more stable after 20 minutes” – what statistical metric is this statement based on?

A t-test was used to determine significance. The text was changed and now reads (Results: SARS-CoV-2 Aerostability Correlates with the Ambient Concentration of Gas Phase Carbon Dioxide, 4th paragraph):

As a result, the Delta VOC in elevated [CO₂(g)] is as aero-stable as the wild type at 500 ppm CO₂ after 5 minutes, resulting in a larger fraction of virus remaining infectious after 20 minutes (p=0.04).

Results, Lines 179-184 and Figure 2C: It isn't completely clear what is being graphed here. Is this the ratio of the raw infectivity data from Fig 2B? Please provide more details on how this analysis was performed. Since there are multiple unpaired data points at each time for each CO₂ level, how were the ratios taken? What infectivity assay was utilized here? Again, the discussion of the data from Figure 2C does not include any statistical analysis to support the statements made in the text.

Yes, the ratio of the infectivity is graphed in Figure 2B. To clarify this, the description of the figure in the caption has been expanded to read as follows:

(C) Relative infectivity of aerosolized Delta variant exposed to increased [CO₂] as a function of time. The ratios were estimated using the raw data in Figure 2B at time points where the infectivity was measured for both [CO₂].

For statistical analysis, a hypothesis test for regression slope of the data in Figure 2B found a $p = 0.00038$. This demonstrates that the relative abundance of infectious SARS-CoV-2 in aerosol in elevated levels of [CO₂] increases over time. The same approach to measurement of infectious aerosolised viral load was used throughout this work, as described in the methods. The following has been added to the text (Results: SARS-CoV-2 Aerostability Correlates with the Ambient Concentration of Gas Phase Carbon Dioxide, 4th paragraph):

A hypothesis test regression of slope for the data in Figure 2C found a $p=0.00038$, consistent with a lower rate of viral decay at elevated [CO_{2(g)}].

Results, Lines 186-190: The authors suggest that the analysis published by van Doremalen may be inappropriate since the decay profile of SARS-CoV-2 measured in the present study appears to be multi-phasic. However, as the authors note later in the manuscript, the methods used in the van Doremalen paper that the authors reference here likely represent only a portion of the decay profile, specifically the slow decay phase. Thus, for this situation, where only a single phase is being examined, expressing the stability as a single half-life seems appropriate, which was the approach taken in the van Doremalen reference. This discussion does not add much to manuscript, and it is recommended to remove this discussion on the van Doremalen analysis and focus on the results of the present study.

We considered that this comparison with the approach taken in the van Doremalan paper (and by others) should not be removed. We do argue that studies that measure the decay rate of aerosolised viruses are largely reporting the decay in the slow phase, before which over 80 to 90% of the virus has already been lost. Our proposal that the decay profile is multiphasic, with the majority of virus being lost at times earlier than conventional techniques can make measurements, is critical to the understanding of disease transmission. The van Doremalen manuscript and others, describe the decay with a single half-life, without an initial rapid loss of viability. Models based on such work which apply these slow decay rates throughout, significantly overestimate the proportion of exhaled virus predicted in the air. In Figure 2D, we show that the decay profile does not follow a single half-life. The discussion provided in lines 186-190 explains why assuming a single half-life is problematic (the conditions in the droplet are in constant flux). Given its importance in the overall theme of the manuscript, we prefer to retain this text.

Results, Lines 203-204: The authors generalize the results with the Delta VOC and state that “CO₂ has a profound effect on the overall aerostability of SARS-CoV-2.” However, it needs to be pointed out that these data are for the Delta VOC, and, given the results presented in Figure 1, may not be applicable to other more recent variants, which may be less sensitive to changes pH. This is a major caveat to the results and precludes the generalization that CO₂ has a “profound” influence on all SARS-CoV-2 variants.

In response to this helpful comment, we have made additional measurements of the effect of CO₂ on the aerostability of other variants of SARS-CoV-2. Given that increasing the [CO₂] necessarily reduces the maximum pH the droplet will reach, this suggests that increasing the [CO₂] ought to decrease the rate of viral decay. Indeed, this was found to be the case where a significant increase in aerostability was observed for the Omicron BA.2 and Beta VOCs when exposed to elevated [CO₂]. Accordingly, we have retained the statement in lines 203-204 referred to in this comment. These results were presented in Figure 2A.

Results, Lines 209-211: Please include references for this sentence. Reference #1 may be appropriate here.

References added (Results: Depending on Variant pH Sensitivity, Ambient [CO₂] and Solute Composition Can Affect Viral Aerostability More than Relative Humidity, 1st paragraph).

Results, Lines 217-226 and Figure 3A: The difference in RH sensitivity in the first 15 seconds between the variants is interesting, and has implications for short range transmission. Were the other variants and MHV studied previously similar to BA.2 across all RH levels, or just below the efflorescence point? If the other variants (excluding Delta) and MHV were similar to BA.2 across all RH levels, this might suggest that the Delta variant is an outlier. Thus, results with Delta may not be generalizable across all SARS-CoV-2 variants.

When compared to the Delta variant, the original strain was found to be much more aerostable across a range of humidities. We previously reported the aerostability of the original strain (Oswin, et al, 2022). There was little viability loss at humidities >50% over 2 minutes. Below the efflorescence point, the decay profiles of all the viruses/variants studied to date in respiratory aerosol have been similar. The reviewer is correct that the Delta variant is an outlier in this time region. The text has been adjusted to make this clearer to the reader and now reads (Results: Depending on Variant pH Sensitivity, Ambient [CO₂] and Solute Composition Can Affect Viral Aerostability More than Relative Humidity, 2nd paragraph):

At an RH range between ~50% and 80%, the Delta VOC was found to be less aerostable than the other VOCs studied, rapidly losing over half of its infectivity within 15 seconds of being in the aerosol phase.

Results, Line 233-234: “doubling of the remaining aerosolized viral load” – was this statistically significant?

We have added the following (Results: Depending on Variant pH Sensitivity, Ambient [CO₂] and Solute Composition Can Affect Viral Aerostability More than Relative Humidity, 3rd paragraph):

[...] results in doubling of the remaining aerosolized viral load after 15 s for all RH < 80%. Pooling the three datasets for RH < 80%, we verified these differences were statistically significant by performing a two-sample t-test ($p = 5 \times 10^{-4}$, with 27.5 effective degrees of freedom).

Results, Lines 236-263 and Figure 3C: The data and discussion on the effect of salt concentration and humidity on the infectivity of the Delta variant seem tangential to the main focus of the paper. Recommend moving this to supplemental materials. This would potentially free up space to include details on the methods in the main text.

Both Reviewer 1 and 3 questioned the inclusion of Figure 3C. Figure 3C, and the majority of discussion of it have been moved to the Supplemental Information section.

Results, Lines 272-274: The data from large particle aerosols were used to inform transmission risk for small particle aerosols. Is this appropriate given that Figure 3C demonstrates that particle size may be a factor influencing survival in aerosols?

The differences in aerostability reported in Figure 3C (now Supplemental Figure 1A) are not a product of the droplet size. Rather, they are a function of the fraction of the droplet solute that is an inorganic salt crystal. When efflorescence occurs, the virus taken up by the salt crystal that forms is protected and remains infectious longer. This has been presented in our previous studies (Ref. 14, Oswin et al., Ref. 15, Haddrell et

al.) The pH flux due to the loss of bicarbonate is size independent and, thus, we believe the assumption made in lines 272-274 is valid.

Results, Lines 318-321: In order to estimate the probability of infection, wouldn't it be necessary to utilize a dose-response relationship? The infection probability calculation presented appears to follow the form of an exponential dose-response model, except that it appears to be missing a term for the probability of the organism causing an infection?

It is indeed an exponential dose-response model, though no term is missing. The effect on the organism is contained within the notion of an infectious quantum, which is essentially the minimal dose required to generate a new infection in an organism. The effect of viral viability simply changes the effective number of quanta received by a susceptible individual through its dependence on c . We have clarified this in the text as (Results: Risk of Transmission is Highly Affected by Ambient Concentrations of CO₂, 4th paragraph):

This exponential dose-response relationship is essentially a consequence of the independent action hypothesis (43). The number of quanta n describes the number of infectious viral doses received, incorporating the effects of viral viability and ventilation.

Results, Figure 4: In Figure 4C, after ~ 8 hours, the probability of transmission appears to increase by approximately 2% at the higher CO2 level. In Figure 4D, this change appears to be less, with the change in the probability of transmission after 1000 minutes of only ~0.5%? These changes do not seem like a very large effect. Some additional discussion would be helpful here to put these changes in context for the reader.

We refer the reviewer to the scale of these axis. First, the x-axis scale is not 1,000 minutes but is 1,000 seconds (15-20 minutes). Second, the y-axis is on the scale of a few percent, so an estimated increase of ~2% between the transmission probability of the Delta VOC at [CO₂] of 3,000 and 500 ppm approximately doubles the transmission probability. Putting to one side the geometric magnification of risk at the group-level (as in Eq. 3 and e.g. Figure 5), doubling the risk to an individual would be significant. Note that the scale of Fig. 4D is smaller, so an increase of ~0.5% is again a doubling. Highlighted in 4D is that the change is less because of the ventilation so this is consistent with the paradigm that ventilation mitigates the risk of airborne transmission.

We have appended the following to the discussion of this figure to clarify (Results: Risk of Transmission is Highly Affected by Ambient Concentrations of CO₂, 4th paragraph):

The probability of transmission is sensitive to parameters with large uncertainties like the quanta production rate, so the absolute values of transmission probability in Figures 4C and 4D should not be taken literally. Rather, the important point is that the relative difference in probability between low [CO_{2(g)}] and high [CO_{2(g)}] can be striking; even with good ventilation (Figure 4D), the probability of onwards transmission approximately doubles for [CO_{2(g)}] = 3000 ppm over the [CO_{2(g)}] = 500 ppm after ~15 minutes of exposure.

Results, Line 353-354: "[CO2(g)] is estimated to have a profound effect on overall risk". How much does risk increase? Please further discuss the rationale for this statement. There does not appear any quantitative analysis provided to support this statement.

The reviewer makes a good point. We have added the following discussion of Figure 5 (Results: Risk of Transmission is Highly Affected by Ambient Concentrations of CO₂, 5th paragraph):

By contrast, the effect of increased $[\text{CO}_{2(g)}]$ has a profound effect on the overall risk of transmission. Even in well-ventilated classrooms with 10 air changes per hour (in a recirculating system), we see that the time before an expected transmission occurs is approximately halved by raising the $[\text{CO}_{2(g)}]$.

Discussion, Lines 387-394: This seems like a bit of stretch given the data that are presented. No data are presented showing that a CO₂ concentration of 1000 ppm impacts survival of SARS-CoV-2 Delta. The lowest level evaluated above the 500-ppm baseline appears to be 2000 ppm.

The lowest level was 1,800 ppm, but extra measurements have been made to estimate the lower limit; 800 ppm was found to be sufficient to increase viral aerostability. The ability for ambient levels of CO₂, at those commonly reached in moderately crowded spaces (<2,000 ppm), has never been hypothesised, let alone demonstrated, to affect how long the virus remains infectious in the air. By virtue of the new measurements showing 800 ppm is enough to affect viral aerostability, the claims made in lines 387-394 are valid.

Discussion, line 405 and line 426: Please define “moderate increases” and “small increases”. The data only show an increase in the stability of SARS-CoV-2 Delta in aerosol particles at CO₂ levels of 2000 ppm and above.

“Moderate increases” has been defined as “from 500 to 800 ppm”. The phrase “small increases” was replaced with “moderate increases” to improve clarity.

Discussion, line 417-419: This is an interesting observation, and it would be interesting to measure how much CO₂ is produced during nebulization of viral media as is done in some of the referenced studies.

We reported this in our previous study (Ref 14). From the reference, the $[\text{CO}_2]$ resulting from nebulizing mimics of respiratory fluid into a confined space are shown below:

The amount of CO₂ increase for each study will be different, and a function of many factors (such as nebulization time and rate, drum volume, sampling technique, etc). This relationship has been entirely overlooked and not considered in drum studies. It is also consequential that the only reported $[\text{CO}_2]$ in a confined space in which respiratory fluids were nebulized reached levels far greater than those required to improve viral aerostability (+300 ppm above baseline, see above figure) (Ref 14).

Discussion, line 432: The summary of the triphasic decay profile is useful and a good summary of the process that occurs during particle equilibration.

We appreciate the reviewer's comment.

Discussion, line 484-485: "Moreover, the rate of loss in the Slow Decay Phase is highly [CO₂(g)] dependent". In examining Figure 2B, if the rate of the slow decay is estimated as the slope of the line between the 20- and 40-minute time points, the slopes do not appear to differ greatly between the 500 and 3000 ppm CO₂ levels, which would seem to contradict this statement. However, again, a lack of statistical analysis precludes ready interpretation and evaluation of this statement.

Based on the slopes in Figure 2B, the half-life between 20 and 40 minutes are $\sim 14 \pm 1$ minutes at 500 ppm, and $\sim 82 \pm 2$ minutes at 3,000 ppm (note the small errors on these measures, obtained by propagating errors using the standard technique). The half-life is a slope in log-linear space (as in Figure 2B), so the difference in slopes may appear small on a log-log axis as in Figure 2B. However, the difference in decay time is clearly visible even within Fig. 2B: one can draw a straight line through the error bars for 3000 ppm, but there is a stark *relative* change in infectivity for 500 ppm. The only reason the *absolute* change is small for 500ppm, is because much of the infectivity has already decayed by 20 minutes. The statistical evidence for this qualitative analysis is entirely contained within the error bars of the measurements given above, and no further testing is necessary.

A figure was added to the Supplemental Information (SF7) showing the change in half-life over time, and reference to the figure was added to the text. The text now reads (Discussion: SARS-CoV-2 has a Triphasic Decay Profile, 4th paragraph):

The data collected using closed systems, such as a Goldberg drum, largely miss the Lag and Dynamic phases (perhaps catching the tail end of the Dynamic Phase), and measure primarily in the Slow Decay Phase. Moreover, the rate of loss in the Slow Decay Phase is highly [CO₂(g)] dependent (Figures 2A, 2B, 2D, 3B and Supplemental Figure 7).

Reviewer #2 (Remarks to the Author):

This manuscript provides new data on changes to the aerostability of SARS-CoV-2 aerosol particles due to acidification from dissolution of CO₂. This data represents a significant advancement in our understanding of the transmission of SARS-CoV-2 and potentially many other respiratory viruses. Overall, I find the work to be of high quality and well written. I have a few minor issues that I would like to see addressed prior to publication.

Issues

1. There should be more of a caveat on the issue of particle size. When performing the Well's Riley models, the authors make the assumption that the decay observed for the relatively large aerosol particles (17-25 micrometers) in this study would be extensible to smaller particles, such as those that might be produced in the bronchi or larynx. I think this is overstated. There will be size-dependent impacts due the different surface to volume ratios of the smaller particles that may change the rates fairly significantly. While I think the assumption for the purpose of making a point with the Wells Riley model is acceptable, there should be some additional attention paid to the issue of size-dependence.

The magnitude of the potential pH increase is a function of the starting bicarbonate concentration and [CO₂] in the room and is, thus, size independent. Any pH decrease due to trace acidic species in the air is, as the reviewer indicates, dependent on droplet size. To address this concern, the following caveat has

been added to the Wells Riley model section (Results: Risk of Transmission is Highly Affected by Ambient Concentrations of CO₂, 1st paragraph):

It should be noted, that the rate of pH reduction will be dependent on both droplet size and total acid content in the air, where smaller droplets will be neutralised at a faster rate. The effect of size was explored in this study to a limited degree, and was found to have a minimal effect within the size range explored (Supplemental Figure 6). The precise degree to which the droplet size affects the rate of neutralization (and subsequent increased viral aerostability) should be measured in the future.

In additional experiments undertaken in response to the reviewer's comments, we have explored the effect of droplet size on the interplay between viral decay and [CO_{2(g)}]. When the volume of the droplet was quartered, no significant effect was observed. The additional experimental data is now included in the Supplementary Information (Supplement Figure 6):

Supplemental Figure 6: Effect of initial droplet size on the interplay between viral decay and elevated [CO_{2(g)}]. Based upon the viral titre per droplet, the initial volume of the "Small" droplets (0.6 infectious viral units per droplet) were approximately a quarter of the "Large" droplets (2.8 infectious viral units per droplet). (a) At 120s, increasing the [CO_{2(g)}] was found to have a similar effect on both droplet sizes. (b) Over time, the effect of the increased [CO_{2(g)}] was found to be similar for both particle sizes.

2. Also, on the topic of aerosol size, the authors mention the initial size of the droplet, but don't report the expected resulting size range after evaporation, which is likely RH dependent. A graph of mean sizes, as a function of RH might be helpful.

The relationships between initial droplet size, composition, humidity and final droplet size have been described in detail in our publication (Ref 14). Rather than including a table, the following clarifying statement has been added to the Methods section (Methods, 2nd paragraph):

The final radius of the droplets will depend on RH, droplet composition and initial size; for the droplets in study, the final radius of the droplets ranged between 5 and 10 μm (14).

3. The authors make the comments that "A survey of the literature found that a large portion of manuscripts do not provide the nebulization period, while none have been found to report the $[\text{CO}_2(\text{g})]$." No articles are cited here, and no indication of how the survey was conducted is included. It might be easier to simply highlight the absence of this data from some key papers, that should be cited, and leave it at that.

This is a good point. The sentence has been changed to read (Discussion: Broader Implications, 3rd paragraph):

The amount of CO_2 produced will depend on the amount of sample nebulized, and has been reported to be greater than the 800 ppm necessary to improve viral aerostability (14). It is notable that of the five publications that have reported the decay rate of the original variant of SARS-CoV-2 (11, 29, 50-52), only two provided enough information to estimate the total mass of sample nebulized, while none reported the $[\text{CO}_2(\text{g})]$.

Reviewer #3 (Remarks to the Author)

Review by Thomas Peter

General Response to the Review by Peter

At the outset, it should first be acknowledged that we are in complete agreement with Peter et al. on the importance of understanding the pH of respiratory aerosol for quantifying the airborne survival and transmission of respiratory pathogens. Peter et al. have developed the ResAM microphysical aerosol model to predict changes in airborne infectivity with droplet size and gas phase composition. We have taken an experimental approach to measure changes in airborne infectivity using a novel technology, providing the opportunity to assess varying microphysical impacts on infectivity. Both approaches can advance our understanding of such a crucial step in the airborne disease transmission pathway. Before our own clear experimental verification of the role of pH and models such as the ResAM model, the mechanism of airborne inactivation was variously and ambiguously attributed to relative humidity, salt concentration, oxidative stress, pH and droplet phase. We can now be certain that aerosol pH is the key factor that must be understood. This will be crucial to avoiding similar scenarios that arose during the COVID-19 pandemic, where the WHO did not acknowledge that SARS-CoV-2 could be transmitted by the airborne route for over 12 months.

The review of Peter highlights the need for continuing dialogue between the experimentally measured changes in infectivity in aerosol from our CELEBS approach and the ResAM model. Peter *et al.* hold the view that it is the condensation of acidic species on the respiratory aerosol that inactivate viruses by pushing

the condensed aerosol phase to acidic conditions. We contend that the loss of the bicarbonate buffering capability of exhaled aerosol, which leaves a CO₂ rich environment in the lung for a low CO₂ ambient concentration, leads to highly alkaline conditions that drive the loss of viral infectivity. It is true that both acidic and alkaline conditions can lead to inactivation, based on bulk phase studies.

While the debate about the importance of acidic and alkaline pH conditions in aerosol continues, it is our assertion that the publication of high quality experimental data against which to benchmark models such as ResAM can provide a paradigm shift in our understanding of disease transmission. We argue that our current study is able to achieve this for the following reasons:

- The dependence of infectivity of SARS-CoV-2 on CO₂ concentration mapped in this study can only be consistent with an aerosol that becomes strongly alkaline at low CO₂.
- The consistency between the stabilities of variants in the bulk phase under alkaline conditions and the aerosol measurements further confirms the importance of aerosol alkalinity.

The consequences of these observations are unique and profound in understanding airborne transmission. The former observation not only mandates the critical messaging around improving room ventilation, but it asserts that urgent work is required to extend studies of airborne survival to other pathogens and to a fuller range of environmental conditions. Without this work, we cannot be certain that rising levels of ambient CO₂ will not have significant implications for rates of airborne disease transmission.

Undoubtedly there is much that remains to be done: this really forms the basis of the Peter review and his critique of our work. The presence of other indoor air pollutants, particularly strongly acidic vapours, must be the focus of subsequent studies. Whatever, these secondary effects do not undermine or negate the significance of the mechanistic experimental work presented here, they will be an extension of it.

This is an interesting manuscript on the physicochemical properties of respiratory aerosol, providing evidence that CO₂-induced changes in particle pH can influence the infectivity of airborne SARS-CoV-2 viruses. The same group has previously shown that particles that mimic expiratory aerosol, albeit with much larger initial size (~50 μm), lose a significant fraction of dissolved bicarbonate as CO₂ upon exhalation, allowing their pH to rise from neutral to alkaline pH 11. To the degree airborne viruses are sensitive to alkaline conditions, this physicochemical change, which occurs within a few minutes after the drops are exposed to the gas phase, may lead to their inactivation. In this new manuscript, the authors generalize this finding by asking whether differences between freshly ventilated rooms (with a few hundred ppm of CO₂) and badly ventilated rooms with stale air (with a few thousand ppm) may influence virus inactivation. They could therefore be an air hygiene argument for good ventilation. They further speculate about how climate change with increasing CO₂ levels could affect the activity of such viruses in the future. The experimental work is based on their state-of-the-art electrodynamic balance apparatus (CELEBS) and uniquely applied to the aerostability of the SARS-CoV-2 Delta and Omicron variants.

Major concerns

While this is all great, I am concerned that this manuscript neglects key limitations of their study: (i) the exclusive focus on the effects of CO₂ in laboratory air, excluding minor trace gases that normally present in indoor air and more relevant for viral activity and transmission; (ii) exclusive use of particles larger than 20 μm (diameter in equilibrium with ambient air) while particles of less than 1 micron to a few micrometers are most relevant for infection and spread of viral respiratory infections; (iii) the use of minimal essential media (MEM or DMEM) as matrix for the experiments, which has a significantly higher salt content than, for example, nasal mucus.

We will address the three major concerns raised by the reviewer point by point:

(i) As discussed above, the primary driver of the pH change of the droplet immediately following exhalation is dictated by the HCO_3^- flux from the droplet via CO_2 . The degree to which the secondary effect of trace acid being added to the droplet offsets this primary effect is unclear, and need of further study.

The ResAM model estimates that this secondary process is dominant. It is notable that the ResAM model has yet to be benchmarked to any experimental data. In our previous study (Ref 15, Haddrell et al, 2023), we attempted to benchmark the ResAM model by exposing the levitated droplets to 50 ppb HNO_3 (see figure below):

[REDACTED]

(Left) Figure 4 from ES&T Article by Peter and coworkers. (E) Inactivation times of SARS-CoV-2 as a function of particle radius under various conditions: indoor air with typical composition (black), depleted in NH_3 to 10 ppt (light blue), enriched to 50 ppb HNO_3 (dark blue), or purified air with both HNO_3 and NH_3 reduced to 20 or 1% of typical indoor values (red). Whiskers show reductions of viral load to 10^{-4} (upper end), 10^{-2} (intersection with line) and $1/e$ (lower end). (F) Mean size distribution of number emission rates of expiratory aerosol particles [$dQ/d\log(R)$] for breathing (solid line), speaking and singing (dotted line), and coughing (dashed line).

(Right) Figure 4b from Haddrell et al, 2023. The effect that changing the acid content of the airflow over the levitated droplets has on virus infectivity of the Delta VOC as a function of pH over time. The vertical line in the Left figure indicates the approximate initial size of the droplets studied in Right with a final size of 5-10 μm depending on RH.

According to the ResAM model, for all gas phase compositions (apart from adding 50 ppb HNO_3) there is no predicted difference in decay rate, and the decay rate is independent of droplet size (in (E), red, light red, light blue and black lines).

The addition of 50 ppb HNO_3 is estimated to have a dramatic increase in viral decay (dark blue). For a 25 micron droplet, 99% of the virus would be inactivated in $\sim 10^5\text{s}$ (~ 27 hours) in normal air, and in about $\sim 10^3\text{s}$ (~ 15 minutes) in air with additional HNO_3 . Meaning, the addition of acidic vapor should lead to a 100 to 1,000 fold increase in the decay rate at this droplet size. Experimentally, the opposite trend has been found (above figure, right). In the absence of additional HNO_3 , $\sim 99\%$ of the virus was inactivated in ~ 40 minutes. The addition of ~ 50 ppb HNO_3 lead to a decrease in viral decay (higher viral infectivity after 40 minutes). This difference in decay is not only in magnitude, but in direction. Clearly, there is a discrepancy between the model prediction and the experimental measurement.

While it is important to consider the magnitude of the secondary effect of acidic vapor reducing the rate of viral decay, our focus in this study is on the first process of bicarbonate partitioning, a process that necessarily happens in all environments on exhalation, independent of the room air composition.

(ii) It remains unclear, even for SARS-CoV-2, what are the most important particle sizes for disease transmission and how this varies between different transmission scenarios. The larger number of smaller particles, as even recorded in our own work on exhaled aerosol (1 μm size, e.g. Interface Focus, 2022, 12, 20210078), do not necessarily counteract the much larger volume of a smaller number of larger particles. Among other factors, this depends on where the disease site is within the respiratory tract. Probing the size segregation of viral load (particularly infectious virus) remains extremely challenging. Many studies (including those by Milton and coworkers, e.g. Clin. Infect. Dis., 2021, 10.1093/cid/ciab691) show marginal differences between aerosol below and above 5 μm size, particularly when it is recognised that the requirements for longer sampling time to sample the smaller numbers of larger droplets are infrequently met (Reid and coworker, Analytical challenges when sampling and characterising exhaled aerosol, Aerosol Sci. Technol., 2022, 56, 160–175.).

Indeed, the rate of viral decay is largely independent of size (Figure 4 of EST Article by Peters and coworkers) in the absence of trace acid according to the ResAM model output. That said, aerosol size may play a role in pH dynamics. Discussion of the interplay between droplet size, pH increase due to bicarbonate evaporation (via CO_2), and subsequent slow reduction after the addition of trace acids in the air has been expanded (as also suggested by reviewer #2). Furthermore, additional experimentation was done to further explore the role of droplet size. Comments to address this concern were added (Results: Risk of Transmission is Highly Affected by Ambient Concentrations of CO_2 , 1st paragraph):

It should be noted, that the rate of pH reduction will be dependent on both droplet size and total acid content in the air, where smaller droplets will be neutralised at a faster rate. The effect of size was explored in this study to a limited degree, and was found to have a minimal effect within the size range explored (Supplemental Figure 6). The precise degree to which the droplet size affects the rate of neutralization (and subsequent increased viral aerostability) should be measured in the future.

It should be noted, that the half-lives reported in this study, where the $[\text{CO}_{2(g)}]$ was set to 3,000 ppm, were found to be similar to those reported on the smaller droplets studied using rotating drums (where the $[\text{CO}_{2(g)}]$ is also elevated (Supplemental Figure 7)). This similarity in decay rates highly suggests that the processes explored in this study are directly applicable to those in the smaller size region.

(iii) The concentration of every solute within saliva, lung fluid and nasal mucus is highly variable between individuals, and even in the same person at different times. The concentration of salts in MEM is well within the range of that reported for lung fluid and saliva, and nasal mucus. For example, the $[\text{Na}^+]$ in MEM is 148 mmol/L; salivary sodium levels have been reported between 27 and 217.3 mmol/L (doi: 10.4103/0976-9668.117006).

The reviewer makes the assertion that the salt concentration will have a dramatic effect on viral aerostability. However, the ResAM model output shows that the viral decay rate is identical in pure NaCl droplets, synthetic lung fluid (SLF) droplets and SLF droplets that contain no salt (Figure S15(H), shown below). According to the ResAM model, salt concentration has no effect on viral aerostability.

[REDACTED]

Figure S15(H) from the EST Article y-axis is “99% inactivation time (s)” Color indicates trace (blue) and very low (black) acidic vapor. SLF: Synthetic lung fluid. The time for SARS-CoV-2 to become 99% inactivated was estimated to be nearly identical for droplet containing 100% salt, a mixture of organics and salt, and pure organics.

In our previous study (Ref 15, Haddrell et al, 2023), we demonstrated that the viral decay dynamics in artificial saliva droplets are similar to those in MEM droplets. We hypothesized the explanation for this is that the two formulations have similar concentrations of the component that drives the virus from losing infectivity (specifically HCO_3^-). Additionally, in that study we explored the impact that $[\text{NaCl}]$ had on viral infectivity decay. At an RH of 90%, tripling the $[\text{NaCl}]$ resulted in no significant change in the viral decay profile (figure below).

Figure 5 from Haddrell et al, 2023. Decay curves of SARS-CoV-2 in droplets whose initial composition has been spiked with additional solutes at 90%. ‘n’ indicates the number of individual levitations used to produce the figure.

In combination, these three aspects may make the inactivation of SARS-CoV-2 very different from what is described in this manuscript, and these caveats are not at all addressed in the manuscript.

Based on our responses to points (i)-(iii) above, we feel that the reviewer's assessment is not entirely consistent with the outputs of their ResAM model. We also feel that there is significant ambiguity in the literature surrounding points (i)-(iii) that robust laboratory work is essential to gain significant insights to address these ambiguities. Our study aims to achieve this. We also contend that, while it is important to consider the magnitude of secondary effects of acidic vapor reducing the rate of viral decay, our focus is firmly on the first process of bicarbonate partitioning, a process that necessarily happens in all environments on exhalation, independent of the room air composition.

I continue to think that this will be an interesting paper, but it requires discussing these constraints and correctly acknowledging other literature, and probably belongs into a more specialized journal than Nature Communications. Furthermore, the news value is somewhat lowered because the basic information about CO₂ evaporating from the droplets in their experiments leading to alkalization was recently published (Oswin et al., PNAS 2022).

We disagree with this assessment. This is the first study to propose that ambient levels of CO₂ may affect viral aerostability and the first to actually demonstrate it. All of the findings reported in this study are novel. Showing that CO₂ concentrations as low as 800 ppm affect how long a virus remains infectious in the air changes both how we understand disease transmission and how to characterise indoor air quality.

Specific comments

Below I address the points above in more detail combined with other points line by line.

L. 18-20. Abstract. "Previous studies have shown that a rapid increase in the pH of respiratory aerosols following generation due to changes in the gas-particle partitioning of pH buffering bicarbonate ions and carbon dioxide is a significant factor reducing viral infectivity." No, previously this has been shown (Oswin et al., PNAS, 2022) for MEM/DMEM particles of sizes larger than respiratory particles and under conditions without the trace gases in typical indoor air (e.g., HNO₃, NH₃, HCl, CH₃COOH). Therefore, I think this statement in the abstract is misleading. As Luo et al. (ES&T 2022) show, particles with the characteristics of nasal mucus and sizes more representative of exhaled aerosol particles emitted into typical indoor air turn acidic after typically a minute.

Consistent with our discussion above, we contend that the statement in the abstract remains a primary driver that effects airborne viral stability and transmission. We agree that there are complexities that remain to be addressed (e.g. particle size, condensable acidic gases), but our focus is on the primary process that occurs for particles of all size and in all environments on exhalation.

L. 31-33. Intro. "The inhalation of respiratory aerosol containing ... SARS-CoV-2 has been identified as the dominant route of transmission driving the spread of ... COVID-19 (Duval et al., BMJ, 2022)". In fact, Duval et al. do not mention the "dominant route". Rather, they conclude very carefully: "This rapid systematic review found evidence suggesting that long distance airborne transmission of SARS-CoV-2 might occur in indoor settings such as restaurants, workplaces, and venues for choirs, and identified factors such as insufficient air replacement that probably contributed to transmission. These results strengthen the need for mitigation measures in indoor settings, particularly the use of adequate ventilation."

Reviewer 1 made a similar comment. The phrasing has been changed to now read (Introduction, 1st paragraph):

The inhalation of respiratory aerosol containing the severe acute respiratory syndrome coronavirus-2 (SARS-CoV-2) has been identified as an important route of transmission in the spread of coronavirus disease 2019 (COVID-19) (1).

L. 45-46. *This sentence states that singing and talking reduce the total number of virus containing aerosol droplets. I believe the authors meant to express the opposite.*

Thank you. This is correct. The sentence has been changed, and now reads (Introduction, 2nd paragraph):

For example, changing aerosol production rates (e.g. via lowering the volume of singing or talking) (5, 6), crowding/social distancing policies (7), mask wearing (8), and improved ventilation (9) all reduce the total number of virus-containing aerosol droplets.

L. 68-70. *The authors write: “What is clear is that models (Luo et al., ES&T 2022) and measurements (19, 20) of human exhaled aerosol have both shown consistently that exhaled respiratory aerosol is significantly more alkaline than the fluids within the respiratory tract from which they originate” (where 19, 20 refer to condensed breath condensate measurements). This statement is a distortion of the facts. Rather, Luo et al. show with biophysical modelling for synthetic lung fluid and nasal mucus that exhaled particles of initial radius of 1 μm start out slightly acidic (pH 6.6) in the respiratory tract, undergo very slight alkalization within very the first 0.2 s after exhalation reaching neutral pH (this is the CO₂ effect discussed by the authors), and then acidify to pH 6 after 1 s, pH 5 after 10 s, and finally pH 4 after 100 s. See Figs. 3F and 4A of the work by Luo et al. Their findings are therefore quite the opposite of “significantly more alkaline”. I suggest that the authors double-check the veracity of the claims made based on the existing literature*

There is complexity on which we disagree with the reviewer. Given the complexity, we have decided to remove the reference to their model from this paragraph. Discussion of the experimental studies on exhaled breath condensate remained. In addition, references to saliva (another well studied respiratory fluid with a similar [HCO₃]) and thus similar pH dynamic, was added. The line now reads (Introduction, 4th paragraph):

What is clear is that measurements of human exhaled aerosol (17-19) and saliva (20) have both shown consistently that exhaled respiratory aerosol fluids are significantly more alkaline than the fluids within the respiratory tract from which they originate.

L. 68-70. *The sentence “what is clear is that...” contains a further misunderstanding by assuming that measurements of breath condensate would prove that the aerosol is becoming alkaline. This is not the case. Doctors typically bubble the freshly collected breath condensate with argon for 10 minutes to remove the high amounts of dissolved CO₂ and bicarbonate. Hence, it is not surprising that the pH values reported for breath condensates are alkaline, but this has nothing to do with the evolution of pH of particles exhaled into indoor air.*

While it is true that it is common to deaerate exhaled breath condensate with gases such as argon prior to measuring pH, historically it has not always the case. For example, consider the following (from Pediatric Pulmonology 38:107– 114 (2004)):

Editorial Note: Figure below reprinted with permission from Rosias, P.P.R., Dompeling, E., Dentener, M.A., Pennings, H.J., Hendriks, H.J.E., Van Iersel, M.P.A. and Jöbbsis, Q. (2004), Childhood asthma: Exhaled markers of airway inflammation, asthma control score, and lung function tests. *Pediatr. Pulmonol.*, 38: 107-114. <https://doi.org/10.1002/ppul.20056>. Copyright © 2004 Wiley-Liss, Inc.

Figure 2. Nondeaerated pH of exhaled breath condensate with time (hours) in 2 subjects of the study population.

Clearly the pH of collected exhaled breath condensate increases over time in the absence of argon (in open air, in the presence of trace acidic vapor). Note that the rate may be slower than that in aerosol due to the higher surface to volume ratio of the droplet. The overall magnitude of the pH change of an exhaled breath condensate will be less than that within an aerosol droplet because the exhaled breath condensate contains a large volume of condensed water vapour; when collected, the vapour turns to water and dilutes the aerosol droplets.

A similar trend in pH change is commonly reported in saliva (where a similar underlying pH dynamic occurs) (from European Review for Medical and Pharmacological Sciences, 2014, 18: 2988-2994):

[REDACTED]

Note, that for both of these examples, trace acidic vapor would be present (as they were not measured in purified air). Given that the pH of these bulk solutions will increase following exposure to open air, assuming a similar dynamic in aerosol is valid. The line was clarified and now reads (Introduction, 4th paragraph):

What is clear is that measurements of human exhaled aerosol (17-19) and saliva (20) have both shown consistently that exhaled respiratory aerosol fluids are significantly more alkaline than the fluids within the respiratory tract from which they originate.

L. 79. "Infectivity decay profile" – what is this?

The decay profile of the viral infectivity. To clarify, the sentence now reads (Introduction, 5th paragraph):

This raises three questions: over what ambient concentration range does CO_{2(g)} impact infectivity, to what degree is the viral infectivity decay profile affected by CO_{2(g)}, and how does this change in infectivity affect overall risk of disease transmission?

L. 119. *Figures: Please show at least one example with the individual measurements of the biological replicas instead of mean and standard errors, as only then one can properly judge the uncertainties. How is it possible that uncertainties are so small when determined reductions are only a few 10 %? In many other investigations, including our own, when reductions are 2 orders of magnitude, error bars are much larger.*

A table has been added to the Supplemental Information section to inform the reader of the uncertainties and demonstrate why error bars are so small using this experimental approach

Typically, viral decay in aerosol is measured using a TCID50. A TCID50 uses serial dilution to measure viral infectivity, meaning that it is intrinsically measured on a log scale. As a consequence, this approach will have a fairly high absolute error. In the CELEBS technique, we levitate small populations of droplets and quantify individual infectious virions using the most probable number (MPN) assay. As a consequence we are able to generate highly sensitive and reproducible results.

Figure 1. I found it hard to understand the caption. The first panel is introduced by (A) and then the explanation, the other panels first with the explanation, then by (B) and (C). Please indicate also the size of particles, this information is hard to find. I figure that (B) and (C) are both for RH 90 %? If so, why do infectivities not continue to decrease after 20 minutes?

For clarity, the figure caption has been altered and now reads:

Figure 1 (A) Infectivity of the Delta and Omicron BA.2 VOCs that have been levitated at RHs of 40% and 90%. Data for 90% at times over 100s are offset by 5s to facilitate reader interpretation. (B) and (C) Infectivity of BA.2 and Delta VOCs, respectively, in DMEM 2% FBS bulk solution with pH maintained at 11, measured by (B) cytopathy and (C) immunostaining. Error bars indicate standard error for all figures.

The size of the particles has been moved from the Supplemental Information to the Methods section in order to make it more easily accessible. Figures (B) and (C) are the decay rates measured in the bulk phase. They do continue to decay after 20 minutes; the rate of loss appears slower as the bulk data follows a single log decay profile (as shown in Figure 2D).

L. 134. *The statement that "collectively, the data shown in Figure 1 support the hypothesis that high pH achieved in aerosol" would be more convincing if the figure contained also some results for lower pH.*

The line referenced here by the reviewer is in reference to our previous study (Ref 15) wherein we showed that the high pH of respiratory aerosol drives the loss of viral infectivity. None of the results from that study, nor this study, correlate with the loss of viral infectivity being caused by the aerosol becoming acidic. Moreover, as shown above in the exhaled breath condensate and saliva studies, there is no evidence that respiratory aerosol is acidic in the time scales studied here.

Figure 2. The information that the entire figure refers to RH 90 % is hidden in the ordinate of panel (A).

Text has been added to the caption to clarify the RH.

Figure 2B. Why is the decay of infectivity between 20 and 40 minutes faster for 500 ppm than for 3000 ppm? Can this be fully explained by the increase in aerostability through CO₂?

This dynamic is completely explained through the paradigm presented here. Over time, the elevated levels of CO₂ ensure that the pH of the droplet does not increase so markedly. This is discussed in the manuscript (Results: SARS-CoV-2 Aerostability Correlates with the Ambient Concentration of Gas Phase Carbon Dioxide, 5th paragraph):

However, the decay dynamics are markedly different in the aerosol phase with the rate of loss slowing over time, and slowing more so at higher [CO_{2(g)}]. This is consistent with the hypothesis that the aerosol achieves a high pH before being buffered towards a neutral pH by trace acidic vapor over longer time periods (condensable carbonic acid in this case), regardless of RH. However, the decay rate is never found to increase over the entire time-period studied, suggesting that the pH of the aerosol does not pass through neutral to become acidic during the time period when more than 95% of the viral infectivity is lost.

This dynamic may run counter to the expectations of the reviewer as CO₂ is an acidic vapor and according to the Peter model any acidity ought to increase the rate of viral decay. However, this has not been observed experimentally. This further indicates how the experimental data provided in this study is at odds with the ResAM model over the time scales studied.

Figure 2D. Something is wrong with the ordinate $\ln([A]/[A_0])$. Why do the measurements start at $\exp(4.5)$ then approach a value of 1?

Thank you. The original calculations were normalised to 100 as opposed to 1. The figure has been corrected.

L. 167-169. "With the rate of loss of viral infectivity increased by elevated aerosol alkalinity, any improvement in aerostability of SARS-CoV-2 resulting from elevated [CO_{2(g)}] would be expected to increase over time with a greater tendency towards neutral pH due to the dissolution of carbonic acid." I read this several times but cannot understand.

This sentence has been clarified, and it now reads (Results: SARS-CoV-2 Aerostability Correlates with the Ambient Concentration of Gas Phase Carbon Dioxide, 4th paragraph):

The rate of viral infectivity loss correlates with aerosol alkalinity. Elevated [CO_{2(g)}] limits the amount of bicarbonate leaving the droplet (Eq. 1) and, thus, limits the maximum pH that the droplet will reach. A significant improvement in aerostability of SARS-CoV-2 resulting from elevated [CO_{2(g)}] would be expected to increase over time as the droplet will spend less time at the elevated pH (28).

Figure 3. I did not grasp how the dependence on NaCl concentration is related to the main topic of the manuscript.

Reviewer 1 made the same critique. As such, the impact of NaCl concentration on viral infectivity across a range of humidities has been moved to the Supplemental Information.

L. 254. A “solute salt crystal” – what is this?

The word “solute” has been removed.

L. 258-261. Entangled style of writing, hard to decipher.

This text was edited for clarity, and it now reads (Results: Depending on Variant pH Sensitivity, Ambient [CO₂] and Solute Composition Can Affect Viral Aerostability More than Relative Humidity, 4th paragraph):

When the [NaCl] is doubled, all particles undergo a significant phase change when the RH is below ~75% (Supplemental Figure 1B).

L. 270. Please define “gas phase like”.

The phrase “gas phase like” has been removed as the rest of the sentence provides the necessary context.

L. 405-406. “Moderate increases in [CO₂(g)] affecting the aerostability of SARS-CoV-2 have very broad implications with regards to how all previously published aerovirology experiments should be interpreted.” While I fully agree that the community needs to pay more attention to the composition of the gas phase beyond temperature and relative humidity, I find this an overstatement and think attention also needs to be directed to other air components, such as HNO₃, NH₃, HCl, CH₃COOH, which besides the carbonate buffer introduce chloride buffer, ammonia buffer, etc.

As demonstrated in this work, even small increases of [CO₂] had a significant effect to increase viral aerostability. In the section the reviewer is referring to, the discussion is centred on how drum studies do not measure the [CO₂] in their studies, thus making interpretation difficult. It is a targeted discussion about how the findings of this work demonstrate that past studies, using that experimental approach, should be re-interpreted. The effect of the other acids mentioned by the reviewer were not explored in this study. Moreover, there is no reason to expect that the concentration of these acidic vapours to vary between drum experiments in a similar fashion as CO₂; the source of excess CO₂ in the drum studies is the starting formulation.

L. 585. Something is wrong with the citation.

The citation has been corrected.

REVIEWER COMMENTS

Reviewer #1 (Remarks to the Author):

This manuscript is a revision of a previous manuscript in which the authors explored the importance of a novel pathway potentially involved in the inactivation of SARS-CoV-2 in aerosol particles related to the concentration of CO₂ in the air and pH changes within a particle. The revised manuscript is a substantial improvement over the initial draft. The inclusion of an additional lower CO₂ level better supports some of the authors' assertions, and the addition of a range of statistical analyses improve the quality of the manuscript. Overall, the revised manuscript is well written, clearly articulating the rationale for the study and importance of the results and should be acceptable for publication. However, I did have a few additional comments, mainly in the Results section, that I feel require some additional clarification prior to publication. I have also included some minor comments on the other sections for the authors' consideration.

Abstract – No major comments. Should the 1800 ppm level mentioned in the text be lowered to 800 ppm given the additional experiments that were performed and added to the manuscript?

Introduction – The revised introduction reads very clearly and provides a good summary of previous work and the basis for the present study. No additional major comments, but please consider adding a reference for the statement on lines 60-61 regarding the biocarbonate content of respiratory fluids.

Methods – The revised methods are a nice improvement over the original manuscript with the addition of the text regarding the electrodynamic balance. A few minor comments for consideration:

-Line 112: It is stated that the size of the final particles is between 5 and 10 μm . How was this measured?

-Line 112: There is an extraneous mu (μ) on line 112

-Line 133-135: Consider adding information on the probes used to measure temperature, humidity, and CO₂ concentrations, as well as their location in the system, to the text here.

Results – The study benefits from the additional data at lower CO₂ levels for several VOCs, as well as the statistical analyses performed. However, I do have a few additional comments/questions on this section:

Figure 2A: Were comparisons made between the 3,000 ppm level for the different VOCs? Line 206-207 suggest there weren't any differences, but please consider discussing this comparison in more detail since it is an important addition to the manuscript, especially given the apparent difference in pH sensitivity between the Delta and Omicron isolates shown in Figure 1. Given the lower of pH sensitivity of Omicron, wouldn't it be expected that there would be less of a difference measured between 500 and 3000 ppm relative to Delta?

Also, in 2A the infectivity data for Omicron at 500ppm and 90% RH appear to be different than those shown in Figure 1A (reading from the figures, ~65% in 1A vs ~50% in 2A) – were these separate sets of tests or the same tests? If the tests in Figure 2A were a separate set of tests, was there still a similar significant difference between the infectivity measured for Delta and Omicron at 500 ppm as was shown in Figure 1A?

Lines 340-342 states "the rapid early decay in infectivity of aerosolized SARS-CoV-2 we report here, as well as previously (14), appears to contradict the consensus opinion that airborne transmission prevails as the dominant mode of transmission." Are the authors suggesting airborne

transmission is not the primary route of transmission? While this initial rapid loss certainly lowers the potential for transmission over longer distances, I do not agree that it alone supports the idea that airborne transmission is not important or the dominant mode of transmission (the dominant mode of transmission is still unclear in the reviewer's opinion). Substantial additional evidence would be needed to support this claim. Please expand upon what the authors are proposing here.

No additional major comments for this section, but please consider the few minor comments below:

Lines 152-153: The sentence beginning with "At 2 minutes, we observed..." is a bit unclear. Is the comparison being reported between the Delta and BA.2 VOCs at the 2 minute timepoint or is comparing the change between the 2 minute timepoint and the previous timepoint? It states the change is 26 +/- 6% at 2 minutes – if this is comparing the levels of infectivity for the two variants at the 2 minute timepoint, this value does not appear to match what is shown in Figure 2A, where the difference appears to be approximately 10-15% (red and black squares at 2 minutes).

Lines 193: instead of saying "only the Delta VOC was used" consider revising to "the majority of testing was conducted with the Delta VOC" since additional data were added with Beta and Omicron.

Figure 2B: Please add the RH that these tests were performed to the caption.

Lines 228-243: This is a good summary of the results.

Figure 3B: Consider adding some discussion of the results at 90% to the text, especially since it differs from the other RH levels. This seems to be missing.

Lines 307-315: Please consider moving this text to Supplemental materials, as it is not particularly relevant to the main text.

Lines 380-390: In the modeling that was done, as the dose (i.e. cVt) an individual received increased, what was assumed about how the probability of infection changed (i.e. what dose-infectivity relationship was utilized)? Wouldn't some assumption on the dose-infectivity relationship be needed to assess how the probability of infection increases as the amount of virus in an environment increases?

Discussion – Overall, the discussion is reasonable. However, please consider adding some text to the Discussion section regarding any differences (or lack thereof, depending on the result) in the pH sensitivity of the different variants of SARS-CoV-2 and the potential implications this could have for transmission of one versus another in different indoor environments with differing CO2 levels. This is generally discussed already, but some additional specific discussion related to the results of the present study should be included.

Reviewer #2 (Remarks to the Author):

My concerns regarding this manuscript have been more than adequately addressed, therefore I recommend publication.

Ambient Carbon Dioxide Concentration Correlates with SARS-CoV-2 Aerostability and Infection Risk

by Haddrell et al.

Review of revised manuscript by Thomas Peter

I thank the authors for the revisions of their manuscript (hereafter termed Haddrell et al., 2024). In the following, my original comments are in *black italics*, your response is in **black bold**, and my new response is in blue roman font.

Reviewer #3 (Remarks to the Author) – Review by Thomas Peter

General Response to the Review by Peter

... The review of Peter highlights the need for continuing dialogue between the experimentally measured changes in infectivity in aerosol from our CELEBS approach and the ResAM model. Peter *et al.* hold the view that it is the condensation of acidic species on the respiratory aerosol that inactivate viruses by pushing the condensed aerosol phase to acidic conditions. We contend that the loss of the bicarbonate buffering capability of exhaled aerosol, which leaves a CO₂ rich environment in the lung for a low CO₂ ambient concentration, leads to highly alkaline conditions that drive the loss of viral infectivity. It is true that both acidic and alkaline conditions can lead to inactivation, based on bulk phase studies.

While the debate about the importance of acidic and alkaline pH conditions in aerosol continues, it is our assertion that the publication of high quality experimental data against which to benchmark models such as ResAM can provide a paradigm shift in our understanding of disease transmission. We argue that our current study is able to achieve this for the following reasons:

- **The dependence of infectivity of SARS-CoV-2 on CO₂ concentration mapped in this study can only be consistent with an aerosol that becomes strongly alkaline at low CO₂.**

I do not know from where Haddrell et al. take the confidence to make such a bold statement: “can *only* be consistent with an aerosol that becomes strongly alkaline at low CO₂.” While I cannot properly explain the inactivation observed in their measurements, I am sure that it is not caused by alkaline pH. See below.

- **The consistency between the stabilities of variants in the bulk phase under alkaline conditions and the aerosol measurements further confirms the importance of aerosol alkalinity.**

I do not understand this argument. Why should the consistency between the stabilities of variants confirm the importance of aerosol alkalinity?

The consequences of these observations are unique and profound in understanding airborne transmission. The former observation not only mandates the critical messaging around improving room ventilation, but it asserts that urgent work is required to extend studies of airborne survival to other pathogens and to a fuller range of environmental conditions. Without this work, we cannot be certain that rising levels of ambient CO₂ will not have significant implications for rates of airborne disease transmission.

Undoubtedly there is much that remains to be done: this really forms the basis of the Peter review and his critique of our work. The presence of other indoor air pollutants, particularly strongly acidic vapours, must be the focus of subsequent studies. Whatever, these secondary effects do not undermine or negate the significance of the mechanistic experimental work presented here, they will be an extension of it.

This is an interesting manuscript on the physicochemical properties of respiratory aerosol, providing evidence that CO₂-induced changes in particle pH can influence the infectivity of airborne SARS-CoV-2 viruses. The same

group has previously shown that particles that mimic expiratory aerosol, albeit with much larger initial size (~50 μm), lose a significant fraction of dissolved bicarbonate as CO₂ upon exhalation, allowing their pH to rise from neutral to alkaline pH 11. To the degree airborne viruses are sensitive to alkaline conditions, this physicochemical change, which occurs within a few minutes after the drops are exposed to the gas phase, may lead to their inactivation. In this new manuscript, the authors generalize this finding by asking whether differences between freshly ventilated rooms (with a few hundred ppm of CO₂) and badly ventilated rooms with stale air (with a few thousand ppm) may influence virus inactivation. They could therefore be an air hygiene argument for good ventilation. They further speculate about how climate change with increasing CO₂ levels could affect the activity of such viruses in the future. The experimental work is based on their state-of-the-art electrodynamic balance apparatus (CELEBS) and uniquely applied to the aerostability of the SARS-CoV-2 Delta and Omicron variants.

Major concerns

While this is all great, I am concerned that this manuscript neglects key limitations of their study: (i) the exclusive focus on the effects of CO₂ in laboratory air, excluding minor trace gases that normally present in indoor air and more relevant for viral activity and transmission; (ii) exclusive use of particles larger than 20 μm (diameter in equilibrium with ambient air) while particles of less than 1 micron to a few micrometers are most relevant for infection and spread of viral respiratory infections; (iii) the use of minimal essential media (MEM or DMEM) as matrix for the experiments, which has a significantly higher salt content than, for example, nasal mucus.

We will address the three major concerns raised by the reviewer point by point:

(i) As discussed above, the primary driver of the pH change of the droplet immediately following exhalation is dictated by the HCO₃⁻ flux from the droplet via CO₂. The degree to which the secondary effect of trace acid being added to the droplet offsets this primary effect is unclear, and need of further study.

I would argue differently. Of course, I agree that the loss of inorganic carbon by degassing immediately after exhalation (mainly $H^+ + HCO_3^- \rightarrow H_2O + CO_2 \uparrow$) leads to alkaline conditions. However, in typical indoor air, these alkaline conditions only last for between 1 s and a few minutes depending on particle size (Klein et al., 2022), and in any case stays below pH 10. After the initial phase, the particles readily take up acidic gases from the ambient air (notably HNO₃), which causes the pH to drop to values around 3 in typical indoor air. Therefore, I do not believe that the effect of trace acids being taken up by the droplets is in any way “secondary”, but this is the main effect, dwarfing the inorganic carbon effects (because carbonic acid is so weak).

The ResAM model estimates that this secondary process is dominant. It is notable that the ResAM model has yet to be benchmarked to any experimental data. In our previous study (Ref 15, Haddrell et al, 2023), we attempted to benchmark the ResAM model by exposing the levitated droplets to 50 ppb HNO₃ (see figure below):

[REDACTED]

Figure A. (Left) Figure 4 from ES&T Article by Peter and coworkers. (E) Inactivation times of SARS-CoV-2 as a function of particle radius under various conditions: indoor air with typical composition (black), depleted in NH₃ to 10 ppt (light blue), enriched to 50 ppb HNO₃ (dark blue), or purified air with both HNO₃ and NH₃ reduced to 20 or 1% of typical indoor values (red)... (F) Mean size distribution of number emission rates of expiratory aerosol particles [dQ/dlog(R)] for breathing (solid line), speaking and singing (dotted line), and coughing (dashed line). (Right) Figure 4b from Haddrell et al. (2023). The effect that changing the acid content of the airflow over the levitated droplets has on virus infectivity of the Delta VOC as a function of pH over time. The vertical line in the Left figure indicates the approximate initial size of the droplets studied in Right with a final size of 5-10 μm depending on RH.

According to the ResAM model, for all gas phase compositions (apart from adding 50 ppb HNO₃) there is no predicted difference in decay rate, and the decay rate is independent of droplet size (in (E), red, light red, light blue and black lines).

The addition of 50 ppb HNO_3 is estimated to have a dramatic increase in viral decay (dark blue). For a 25 micron droplet, 99% of the virus would be inactivated in $\sim 10^5\text{s}$ (~ 27 hours) in normal air, and in about $\sim 10^3\text{s}$ (~ 15 minutes) in air with additional HNO_3 . Meaning, the addition of acidic vapor should lead to a 100 to 1,000 fold increase in the decay rate at this droplet size. Experimentally, the opposite trend has been found (above figure, right). In the absence of additional HNO_3 , $\sim 99\%$ of the virus was inactivated in ~ 40 minutes. The addition of ~ 50 ppb HNO_3 lead to a decrease in viral decay (higher viral infectivity after 40 minutes). This difference in decay is not only in magnitude, but in direction. Clearly, there is a discrepancy between the model prediction and the experimental measurement.

This discrepancy is indeed perplexing at first glance. But the mystery can be solved by using the Pitzer ion interaction model to calculate the vapor pressure of 1.9×10^{-5} M aqueous HNO_3 used by Haddrell et al. (2023). See Figure B. Such a solution has an HNO_3 vapor pressure corresponding to 2×10^{-7} ppbv at standard conditions (STP) instead of the 50 ppbv intended by Haddrell et al. (2023). I am not sure how this error of 8 orders of magnitude could have happened, but on this basis it is clear that their experiments could not possibly have shown any effect of the HNO_3 . I hope and expect that the authors will write a corrigendum for their paper Haddrell et al. (2023).

Figure B. Nitric acid vapor pressure over aqueous HNO_3 solutions with molalities of 10^{-5} to 0.4 mol HNO_3 per kg of H_2O calculated by means of the Pitzer ion interaction model. The precise parameterization is taken from Carslaw et al. (1995), see Section III Eqs. (18) and (20). Model results (black line) are shown as HNO_3 mixing ratio at STP. Green arrows: aqueous HNO_3 solution with 0.4 mol/kg has $p_{\text{HNO}_3} = 5 \times 10^{-5}$ hPa, i.e. 50 ppbv at STP, as discussed by Luo et al. (2023). Red arrows: aqueous HNO_3 solution with 1.9×10^{-5} M (which is almost the same as 1.9×10^{-5} mol $\text{HNO}_3/\text{kg H}_2\text{O}$) has $p_{\text{HNO}_3} = 2 \times 10^{-13}$ hPa, i.e. 2×10^{-7} ppbv at STP, as used by Haddrell et al. (2023) in their acidified experiments.

Reference: Carslaw, K.S., S. L. Clegg, and P. Brimblecombe, A Thermodynamic Model of the System $\text{HCl-HNO}_3\text{-H}_2\text{SO}_4\text{-H}_2\text{O}$, Including Solubilities of HBr , from <200 to 328 K, *J. Phys. Chem.*, 99, 11557-11574 11557, 1995

I am not sure about the argument, that the (small) “difference in decay is not only in magnitude, but in direction”. I think that the “direction” is subject to low statistical significance, as the error bars in Figure A (right) seem to suggest. Could it be that the differences are statistically insignificant and that the results are actually equal, as they should be given the tiny amounts of HNO_3 applied?

Clearly, there is a discrepancy between the model prediction and the experimental measurement.

I continue to think that the thermodynamics and kinetics as modeled by ResAM are sound. In contrast, the experimental work suffers from wrong assumptions or interpretations. Furthermore, I wonder why the right panel of Figure A classifies the red bars as “ambient HNO_3 ”. Most experimental work on viruses in aerosols does not specify how the air used in the experiments was exactly processed and what its composition is. Also the current manuscript, similar to other papers of Reid group, does not provide this information. One needs to go back to Fernandez et al. (2019), who introduced the CELEBS technique. This paper states that the air flow is “delivered by an air purifier (Precision Air Compressor, Peak Scientific, UK)”, but how the air is purified and how much trace gas it then still contains remains unclear and cannot be easily obtained from the webpage of Precision Air. Given the importance of the issue it would be important to explicitly state the air composition.

While it is important to consider the magnitude of the secondary effect of acidic vapor reducing the rate of viral decay, our focus in this study is on the first process of bicarbonate partitioning, a process that necessarily happens in all environments on exhalation, independent of the room air composition.

In the light of the above, I remain of the opinion that the partitioning of bicarbonate is of lesser importance than the uptake of other gases from the air.

(ii) It remains unclear, even for SARS-CoV-2, what are the most important particle sizes for disease transmission and how this varies between different transmission scenarios. The larger number of smaller particles, as even recorded in our own work on exhaled aerosol (1 μm size, e.g. Interface Focus, 2022, 12, 20210078), do not necessarily counteract the much larger volume of a smaller number of larger particles. Among other factors, this depends on where the disease site is within the respiratory tract. Probing the size segregation of viral load (particularly infectious virus) remains extremely challenging. Many studies (including those by Milton and coworkers, e.g. Clin. Infect. Dis., 2021, 10.1093/cid/ciab691) show marginal differences between aerosol below and above 5 μm size, particularly when it is recognised that the requirements for longer sampling time to sample the smaller numbers of larger droplets are infrequently met (Reid and coworker, Analytical challenges when sampling and characterising exhaled aerosol, Aerosol Sci. Technol., 2022, 56, 160–175.).

It is true that the larger number of smaller particles does not necessarily compensate for the much larger volume of a smaller number of larger particles. However, in the cited study by Milton and coworkers, aerosols were only collected in 2 size fractions, namely $>5 \mu\text{m}$ and $\leq 5 \mu\text{m}$ in diameter (coarse and fine), which does not help to decide whether the particles studied by Haddrell et al. are larger than respiratory aerosols or not. Other measurements indicate that the aerosol particles involved in the COVID transmission chain are significantly smaller than the particles used by Haddrell. For example, Figure C shows the concentrations of SARS-CoV-2 genetic material in the air for 14 particle size ranges (spanning 10 nm to 32 μm aerodynamic diameter) at different sampling locations, providing one of the largest available data sets of airborne size-fractionated SARS-CoV-2 RNA. The particles used by Haddrell et al. with an original diameter of 50 μm shrink within 1 s (due to H_2O loss) to $\sim 14 \mu\text{m}$ and are indicated by the red arrow in Figure C. This suggests that their laboratory particles are larger than the vast majority of the particles carrying viral RNA. This may be related in part to the fact that such large particles settle quickly due to gravity, reducing their importance for airborne transmission. Particles with a diameter of 14 μm take 2 minutes to fall 1 m, rapidly removing them from indoor air and decreasing their importance in the COVID infection chain. I suspect that this is the reason why no larger particles were found in these hospital measurements, although measurements extended up to 32 μm .

[REDACTED]

Figure C. Size distributions of airborne particles containing SARS-CoV-2 genetic material sampled in hospital and home care environments occupied by COVID-19 positive subjects (Cvitešić Kušan et al., 2023). The measurements were performed with a nanoMOUDI 122R cascade impactor separating particles in 14 size fractions: 0.010–0.018, 0.018–0.032, 0.032–0.056, 0.056–0.100, 0.100–0.180, 0.180–0.320, 0.320–0.560, 0.560–1.0, 1.0–1.8, 1.8–3.2, 3.2–5.6, 5.6–10, 10–18, 18–32 μm . The investigation was conducted in the presence of infected individuals with mild symptoms of COVID-19 including fever, coughing and sneezing. The red arrow shows the size of the particles used by Haddrell et al. after initial loss of H_2O is taken into account ($\sim 14 \mu\text{m}$).

Reference: Cvitešić Kušan, A., J. Baranašić, S. Frka, T. Lucijanić, A. Šribar, J. Knežević, G. Buonanno, L. Stabile, The size distribution of SARS-CoV-2 genetic material in airborne particles sampled in hospital and home care environments occupied by COVID-19 positive subjects, Sci. Total Environ., 892, DOI10.1016/j.scitotenv.2023.164642, 2023.

Indeed, the rate of viral decay is largely independent of size (Figure 4 of EST Article by Peters and coworkers) in the absence of trace acid according to the ResAM model output. That said, aerosol size may play a role in pH dynamics. Discussion of the interplay between droplet size, pH increase due to bicarbonate evaporation (via CO_2), and subsequent slow reduction after the addition of trace acids in the air has been expanded (as also suggested by reviewer #2). Furthermore, additional experimentation was done to further explore the role of droplet size. Comments to address this concern were added (Results: Risk of Transmission is Highly Affected by Ambient Concentrations of CO_2 , 1st paragraph):

It should be noted, that the rate of pH reduction will be dependent on both droplet size and total acid content in the air, where smaller droplets will be neutralised at a faster rate. The effect of size was

explored in this study to a limited degree, and was found to have a minimal effect within the size range explored (Supplemental Figure 6). The precise degree to which the droplet size affects the rate of neutralization (and subsequent increased viral aerostability) should be measured in the future.

When the authors write that “it should be noted that the rate of pH reduction will be dependent on both droplet size and the total acid content in the air, with smaller droplets being neutralized faster”, this is not clear and could be misleading. It ignores the results of Luo et al. (2022) and Klein et al. (2022), who unmistakably demonstrated that the aerosol in typical indoor air does not just “neutralize”, but becomes acidic with a pH < 4 (within 3 minutes for particles with an initial diameter of 4 μm and within half a day for particles with an initial diameter of 50 μm).

It should be noted, that the half-lives reported in this study, where the [CO_{2(g)}] was set to 3,000 ppm, were found to be similar to those reported on the smaller droplets studied using rotating drums (where the [CO_{2(g)}] is also elevated (Supplemental Figure 7)). This similarity in decay rates highly suggests that the processes explored in this study are directly applicable to those in the smaller size region.

Supplementary Figure 7 does not specify the droplets size and the air composition, so I cannot judge how similar the decay rates are or not. I do not want to deny the scientific value of the measurements performed by Haddrell et al. (2024). These are very good experiments that clarify a not well understood and important aspect of very large expiratory particles in the COVID infection chain. However, the caveats that apply to these experiments are not clearly stated, and main claim that the observed loss of infectivity is caused by the initial CO₂ loss remains unfounded. I continue to think that this claim is incorrect.

(iii) The concentration of every solute within saliva, lung fluid and nasal mucus is highly variable between individuals, and even in the same person at different times. The concentration of salts in MEM is well within the range of that reported for lung fluid and saliva, and nasal mucus. For example, the [Na⁺] in MEM is 148 mmol/L; salivary sodium levels have been reported between 27 and 217.3 mmol/L (doi: 10.4103/0976-9668.117006).

The reviewer makes the assertion that the salt concentration will have a dramatic effect on viral aerostability. However, the ResAM model output shows that the viral decay rate is identical in pure NaCl droplets, synthetic lung fluid (SLF) droplets and SLF droplets that contain no salt (Figure S15(H), shown below). According to the ResAM model, salt concentration has no effect on viral aerostability.

...

I accept the answer to point (iii). The authors argue convincingly that the salt content is probably only of minor importance. I withdraw this criticism.

In combination, these three aspects may make the inactivation of SARS-CoV-2 very different from what is described in this manuscript, and these caveats are not at all addressed in the manuscript.

Based on our responses to points (i)-(iii) above, we feel that the reviewer’s assessment is not entirely consistent with the outputs of their ResAM model.

The authors’ response to point (i), i.e. the exclusive focus on the effects of CO₂ in laboratory air, excluding minor trace gases, is based on an error in calculating the concentration of a solution in equilibrium with 50 ppbv HNO₃ (at STP). Their response to point (ii), i.e. the exclusive use of particles larger than relevant for infection and spread of viral respiratory infections, ignores evidence from cascade impactor measurements showing that the size of their particles (~ 14 μm in diameter after H₂O equilibrated with the gas phase) is at the extreme end of the observed size spectrum, as is to be expected at the sedimentation speed of 0.5

m/minute. It is only in relation to point (iii), i.e. the significantly higher salt content than nasal mucus, that I accept their refutation. I do not see in what sense my assessment is not entirely consistent with the outputs of ResAM (except for point (iii) that I withdraw).

We also feel that there is significant ambiguity in the literature surrounding points (i)-(iii) that robust laboratory work is essential to gain significant insights to address these ambiguities. Our study aims to achieve this. We also contend that, while it is important to consider the magnitude of secondary effects of acidic vapor reducing the rate of viral decay, our focus is firmly on the first process of bicarbonate partitioning, a process that necessarily happens in all environments on exhalation, independent of the room air composition.

I fully support the plea for solid laboratory work. However, this must go hand in hand with an appropriate and careful interpretation of the measured data.

Therefore, I wondered on what the authors base their claim that the initial CO₂ loss causes the measured SARS-CoV-2 inactivation. Since they distrust our model, I tried to rely only on their own measurements, which show that even the most alkaline conditions they studied in the cell culture medium DMEM cannot explain their observed inactivation of SARS-CoV-2 in aerosol particles of 50 μm in diameter:

[REDACTED]

Figure D. Measurements by the Reid group of developing alkalinity in the cell culture medium DMEM and the inactivation SARS-CoV-2. (i) The pH changes that DMEM underwent when exposed to open air. Identical to Fig. 5B of Oswin et al. (2022). DMEM was left in an open Petri dish (or a 50-mL tube) and the initial pH (blue bar) and the pH after 20 minutes (orange bar) were measured. The same measurement was carried out using thin layers of DMEM that were allowed to evaporate to 10 % of their original volume over the course of 24 h. **(ii) Infectivity loss of SARS-CoV-2 after 20 minutes of high pH exposure in DMEM bulk solutions.** Identical to Fig. S8 of Oswin et al. (2022). **(iii) Infectivity of the Delta variant ($\ln([A]/[A_0])$) as a function of time in 50-μm aerosol particles (black lines) and in bulk solutions (green solid and read dashed lines).** Identical to Fig. 2D of the current submission by Haddrell et al. (2024) with the addition of the red dashed line for pH9.5.

The red text and red lines in Fig. D mark the highest pH observed by the authors after allowing a thin layer of DMEM to evaporate to 10% of its original volume, which equilibrated with the air in the laboratory (~ 500 ppm CO₂) over the course of 24 h (labeled “fully evaporated” in panel i). At this pH, the infectivity in the bulk reduces to about 77 % (panel ii). If the inactivation in aerosol were a 1st order process, this would result in the red dashed inactivation curve in panel (iii), which is far too flat to explain the observed inactivation in 50-μm aerosol particles. Therefore, I do not see how the authors’ claims could be sustained.

Using ResAM, we attempted to model Oswin et al.’s pH values measured in Petri dish experiments and compare them with the inactivation experiments using aerosol particles with initial diameter of 50 μm described by Haddrell et al. (2024), see Fig. E. The pH modeled by ResAM (left panel) in large DMEM drops with initial radius of 1.5 mm (which like the thin DMEM layer on the Petri dish also shrinks to 10 % of its original volume) agrees reasonably well with the measured pH values. However, using the same modeled pH evolution, ResAM cannot capture the results of the aerosol measurements by Haddrell et al. (right panel). At best, the model can describe the phase for $t > 20$ min (500 ppm CO₂), but there is no way in which the initial rapid decay could be described. The initial phase would require pH ~ 12, which is impossible to reach in such particles.

Furthermore, if the entire evolution of the active titer shown in the right panel were due to changes in alkalinity, the pH would initially have to be very high to drive the rapid decay and subsequently tend towards

more neutral values to explain the slowing of inactivation. However, it must be clear that the pH of such droplets exposed to clean laboratory air without other trace gases must be a monotonically increasing function of time. There is simply no mechanism that allows the pH value to first be large and then to shrink again. So, while the initial phase of inactivation might be modulated by pH, something other than pH must be the main cause of this rapid inactivation. The authors struggle with this problem themselves and therefore come up with what they call the “Triphasic Viral Aerosol Decay (TVAD)” profile (Fig. 6 of the submitted manuscript). This is a nice sketch but does not explain what is actually happening.

Figure E. ResAM modeling of measurements by the Reid group. (left) Modeled pH. Vertical lines with delimiters: Petri dish measurements of pH in DMEM exposed to laboratory air at $t = 0, 20 \text{ min}$ and 24 h . Lines: model results for a large DMEM droplet with an initial radius of 1.5 mm . Solid line: assuming carbonic acid dissociation equilibria to be established instantaneously. Dashed line: with slow kinetics of the conversion of bicarbonate to carbon dioxide. **(right) Modeled reduction in infective SARS-CoV-2 titer.** Black and red symbols: measurements in Fig. 2D of the present submission (Haddrell et al., 2023). Light blue line: ResAM result of SARS-CoV-2 titer in evaporating drop (shifted along the $\ln([A]/[A_0])$ axis to be comparable with the later phase of the measurements at 500 ppmv CO_2).

I continue to think that this will be an interesting paper, but it requires discussing these constraints and correctly acknowledging other literature, and probably belongs into a more specialized journal than Nature Communications. Furthermore, the news value is somewhat lowered because the basic information about CO₂ evaporating from the droplets in their experiments leading to alkalization was recently published (Oswin et al., PNAS 2022).

We disagree with this assessment. This is the first study to propose that ambient levels of CO₂ may affect viral aerostability and the first to actually demonstrate it. All of the findings reported in this study are novel. Showing that CO₂ concentrations as low as 800 ppm affect how long a virus remains infectious in the air changes both how we understand disease transmission and how to characterise indoor air quality.

No, they help only to understand the behavior of very large DMEM droplets in the absence of other acidic trace gases, which is not the case for transmission in typical indoor air. Our previous results show that IAV and SARS-CoV-2 are emitted in particles with $r \ll 10 \mu\text{m}$ and much less in larger particles, so I conclude that acidification and not alkalization is relevant for the infection pathway. As well there must be another unknown mechanism independent of pH that is responsible for the rapid initial decay observed by Haddrell et al. (2024).

Specific comments. Since I disagree with the authors’ response to my major concerns, I will not go into detail about their replies to my specific comments. I will only point out that their examples of studies on exhaled breath condensate, whether with or without deaeration, also provide no evidence for $\text{pH} > 7.6$, which is not alkaline enough to cause significant inactivation of SARS-CoV-2.

Summary. The authors decided to refute my concerns. However, I still believe that this study investigates the interesting case of inactivation of SARS-CoV-2 in large expectoration particles under clean conditions that are atypical for normal indoor conditions. Unfortunately, I remain unconvinced of the “critical importance” of alkaline pH, of “maintaining low CO₂ concentrations in indoor environments for mitigating disease transmission”, or even of “the increased risks of respiratory pathogen transmission ... as our climate changes”. According to the current state of knowledge, it is an exaggeration to blame the climate-induced rise in CO₂ concentration for a change in the infectivity of airborne pathogens.

In the following, the Reviewer's comments are identified in *italics*, our response in **bold**, and changes made to the manuscript are in normal text and indented; the location of the updated text are identified by the section and paragraph in which they were made.

Reviewer 1

This manuscript is a revision of a previous manuscript in which the authors explored the importance of a novel pathway potentially involved in the inactivation of SARS-CoV-2 in aerosol particles related to the concentration of CO₂ in the air and pH changes within a particle. The revised manuscript is a substantial improvement over the initial draft. The inclusion of an additional lower CO₂ level better supports some of the authors' assertions, and the addition of a range of statistical analyses improve the quality of the manuscript. Overall, the revised manuscript is well written, clearly articulating the rationale for the study and importance of the results and should be acceptable for publication. However, I did have a few additional comments, mainly in the Results section, that I feel require some additional clarification prior to publication. I have also included some minor comments on the other sections for the authors' consideration.

We thank the reviewer for their input.

Abstract – No major comments. Should the 1800 ppm level mentioned in the text be lowered to 800 ppm given the additional experiments that were performed and added to the manuscript?

1,800 has been edited to now read 800.

Introduction – The revised introduction reads very clearly and provides a good summary of previous work and the basis for the present study. No additional major comments, but please consider adding a reference for the statement on lines 60-61 regarding the bicarbonate content of respiratory fluids.

A reference that includes a literature survey of the concentrations of a range of relevant components, including bicarbonate, for both respiratory fluids and growth medium has been added.

Methods – The revised methods are a nice improvement over the original manuscript with the addition of the text regarding the electrodynamic balance. A few minor comments for consideration:

Appreciated.

-Line 112: It is stated that the size of the final particles is between 5 and 10 μm . How was this measured?

These were measured using a comparative kinetic electrodynamic balance. We reported these measurements in the citation accompanying the statement. An additional clarification was added to inform the reader of this. The statement now reads:

The final radius of the droplets will depend on RH, droplet composition and initial size; with regards to the droplet compositions and sizes used in this study, our previous published work has shown the final radius between 5 and 10 μm in our previous work (14).

-Line 112: There is an extraneous mu (μ) on line 112

Removed.

-Line 133-135: Consider adding information on the probes used to measure temperature, humidity, and CO₂ concentrations, as well as their location in the system, to the text here.

Information on the devices used to measure the temperature, humidity and CO₂ concentration was already provided in the Materials and Methods (CELEBS – Airborne Longevity Measurements) section of the Supplemental Information. In the interest of brevity, the decision has been made to leave the information there.

Results – The study benefits from the additional data at lower CO₂ levels for several VOCs, as well as the statistical analyses performed. However, I do have a few additional comments/questions on this section:

Figure 2A: Were comparisons made between the 3,000 ppm level for the different VOCs? Line 206-207 suggest there weren't any differences, but please consider discussing this comparison in more detail since it is an important addition to the manuscript, especially given the apparent difference in pH sensitivity between the Delta and Omicron isolates shown in Figure 1. Given the lower of pH sensitivity of Omicron, wouldn't it be expected that there would be less of a difference measured between 500 and 3000 ppm relative to Delta?

The effect of increasing the [CO₂] from 500 ppm to 3,000 ppm resulted in a significant increase in aerostability of both the Omicron and Beta VOCs. The reviewer is correct that the degree to which the viral infectivity changes due to an increase of [CO₂] will be dependent on the VOC. Commentary discussing this point was added.

The third paragraph in *Results: SARS-CoV-2 Aerostability Correlates with the Ambient Concentration of Gas Phase Carbon Dioxide* has been edited, and now reads:

Increasing the [CO₂(g)] resulted in a significant increase in the aerostability of both the Beta and Omicron BA.2 VOCs at 120s. This suggests that the viral aerostability is dependent on CO₂(g) concentration for all SARS-CoV-2 variants. The increase in infectivity of the Omicron variant due to the elevated [CO₂] whilst similar, was slightly lower (+11.7%) than the Beta (+23.4%) and Delta (+36.8%) VOCs. This is likely a product of the differing pH sensitivities for different variants, wherein the more pH sensitive variants are more sensitive to changes in [CO₂]. It is unlikely a similar increase across variants will occur at all time periods for all variants.

Also, in 2A the infectivity data for Omicron at 500ppm and 90% RH appear to be different than those shown in Figure 1A (reading from the figures, ~65% in 1A vs ~50% in 2A) – were these separate sets of

tests or the same tests? If the tests in Figure 2A were a separate set of tests, was there still a similar significant difference between the infectivity measured for Delta and Omicron at 500 ppm as was shown in Figure 1A?

This was a typographical error with the bars in Figure 2A for 0 and 500 ppm shown as incorrect values. The 500 ppm value now matches the value found in Figure 1A.

Lines 340-342 states “the rapid early decay in infectivity of aerosolized SARS-CoV-2 we report here, as well as previously (14), appears to contradict the consensus opinion that airborne transmission prevails as the dominant mode of transmission.” Are the authors suggesting airborne transmission is not the primary route of transmission? While this initial rapid loss certainly lowers the potential for transmission over longer distances, I do not agree that it alone supports the idea that airborne transmission is not important or the dominant mode of transmission (the dominant mode of transmission is still unclear in the reviewer’s opinion). Substantial additional evidence would be needed to support this claim. Please expand upon what the authors are proposing here.

We agree with the assertion of the reviewer. Our intention in this paragraph was to make it clear that while the decay rates that we are reporting here are faster than those reported using rotating drums, there remains enough infectious virus for airborne transmission with the fast decay leading to a loss of one order of magnitude in infectivity only. In no way are we arguing that airborne transmission is not a major route of transmission. Based upon these comments by the reviewer, we have edited the paragraph to clarify our position.

The second paragraph of *Results: Risk of Transmission is Highly Affected by Ambient Concentrations of CO₂* now reads:

To be clear, the rapid early decay in infectivity of aerosolized SARS-CoV-2 we report here, as well as previously (14), does not contradict the consensus opinion that airborne transmission prevails as the dominant mode of transmission. Our objective here is to demonstrate that the decay dynamics reported in Figures 1A and 2B are actually consistent with this consensus, especially in indoor environments. We therefore focus on the limit of a well-mixed indoor environment using the Wells-Riley framework and using our refined characterization of the infectivity decay rate.

No additional major comments for this section, but please consider the few minor comments below:

Lines 152-153: The sentence beginning with “At 2 minutes, we observed...” is a bit unclear. Is the comparison being reported between the Delta and BA.2 VOCs at the 2 minute timepoint or is comparing the change between the 2 minute timepoint and the previous timepoint? It states the change is 26 +/- 6% at 2 minutes – if this is comparing the levels of infectivity for the two variants at the 2 minute timepoint, this value does not appear to match what is shown in Figure 2A, where the difference appears to be approximately 10-15% (red and black squares at 2 minutes).

The phrasing of the sentence has been adjusted to improve clarity. The 3rd to last sentence of the first paragraph in *Results: The BA.2 Omicron VOC is More Aerostable than the Delta VOC* now reads:

Excluding data points in the transient decay (< 1 minute) to focus on the more gradual decay from 2 minutes onwards, our analysis revealed significant differences in infectivity between the Delta and BA.2 VOCs. At the 2 minute time point, we observed a statistically

significant difference in the infectivity of the Delta and Omicron VOCs of $26 \pm 6\%$ ($p=3 \times 10^{-4}$, with 19.8 effective degrees of freedom via Satterthwaite approximation) at 90% RH.

Regarding the reviewer's comment around the 26% difference in infectivity at 2 minutes. At 2 minutes, the difference between the Omicron BA.2 and Delta infectivity at 90% RH (red and black squares respectively) is 26%, and not "10-15%" (Figure 2A, shown below). No change was made.

Lines 193: instead of saying "only the Delta VOC was used" consider revising to "the majority of testing was conducted with the Delta VOC" since additional data were added with Beta and Omicron.

Good suggestion. The final sentence in the first paragraph in *Results: SARS-CoV-2 Aerostability Correlates with the Ambient Concentration of Gas Phase Carbon Dioxide* now reads:

Thus, in order to explore the effect that $[\text{CO}_2(\text{g})]$ has on viral aerostability across a broad range of conditions and over long time periods, the majority of the testing was conducted with the Delta VOC as it afforded a much higher measurement throughput.

Figure 2B: Please add the RH that these tests were performed to the caption.

RH added. The caption for Figure 2B now reads:

(B) The effect that an elevated concentration of CO_2 has on the decay profile of the Delta VOC and original strain of SARS-CoV-2 at 90% RH. Inset is simply a zoom in of the first 5 minutes of the x-axis. Elevating the $[\text{CO}_2(\text{g})]$ results in a significant difference in overall decay ($p < 0.05$) of the Delta VOC from 2 minutes onward.

Lines 228-243: This is a good summary of the results.

Thank you.

Figure 3B: Consider adding some discussion of the results at 90% to the text, especially since it differs from the other RH levels. This seems to be missing.

The following discussion has been added to the end of the third paragraph in *Results: Depending on Variant pH Sensitivity, Ambient [CO₂] and Solute Composition Can Affect Viral Aerostability More than Relative Humidity*:

At 90% RH, no viral decay was measured (Figures 3A and 3B); droplets injected into an RH of 90% are still evaporating at 15 seconds and, thus, the conditions in the droplet are not sufficiently different to affect viral infectivity.

Lines 307-315: Please consider moving this text to Supplemental materials, as it is not particularly relevant to the main text.

As suggested, the text has been moved to the Supplemental Information: Results section (below Supplemental Figure 1).

Lines 380-390: In the modelling that was done, as the dose (i.e. cVt) an individual received increased, what was assumed about how the probability of infection changed (i.e. what dose-infectivity relationship was utilized)? Wouldn't some assumption on the dose-infectivity relationship be needed to assess how the probability of infection increases as the amount of virus in an environment increases?

The infectious dose required will depend on numerous factors, such as vaccination status, prior infection history, etc.). Thus, the Wells-Riley model estimates the relative risk assuming exposure to a given number of infectious units, or “quanta”. If a given variant was found to have a higher infectious dose, the quanta value would simply be increased. The aim of the Wells-Riley model in this study is to demonstrate the change in relative risk that results from changing a single parameter ([CO₂]).

Discussion – Overall, the discussion is reasonable. However, please consider adding some text to the Discussion section regarding any differences (or lack thereof, depending on the result) in the pH sensitivity of the different variants of SARS-CoV-2 and the potential implications this could have for transmission of one versus another in different indoor environments with differing CO₂ levels. This is generally discussed already, but some additional specific discussion related to the results of the present study should be included.

As suggested earlier, additional discussion into this topic has been now added to the Results section (as suggested above). Given the length/structure of the Discussion section, we feel that reiterating this point is unnecessary. In the interest of brevity, no changes have been made specific for this point.

Reviewer #2:

My concerns regarding this manuscript have been more than adequately addressed, therefore I recommend publication.

Appreciated.

Reviewer #3:

As described in detail in their review, Reviewer #3 has made it clear that the experimental outputs of our study do not align with the ResAM model. We are in agreement with this. We feel that the ResAM model still requires critical evaluation and benchmarking. Thus, only the comments provided by Reviewer #3 that evaluate the manuscript itself, and not how it relates to the ResAM model outputs, will be addressed here.

Page 3, Paragraph 4: "Most experimental work on viruses in aerosols does not specify how the air used in the experiments was exactly processed and what its composition is. Also the current manuscript, similar to other papers of Reid group, does not provide this information. One needs to go back to Fernandez et al. (2019), who introduced the CELEBS technique. This paper states that the air flow is "delivered by an air purifier (Precision Air Compressor, Peak Scientific, UK)", but how the air is purified and how much trace gas it then still contains remains unclear and cannot be easily obtained from the webpage of Precision Air. Given the importance of the issue it would be important to explicitly state the air composition."

An air purifier was not used in this study. As mentioned in the *Materials and Methods* section of the SI, depending on the specific measurement made, either an air compressor or synthetic air was used.

The first sentence in *SI: Materials and Methods (CELEBS-Airborne Longevity and Measurements)* has been edited and reads:

The sources of compressed air used in the study were either CO₂-free compressed air (for experiments where the concentration of CO₂ was set to 0 ppm) or compressed laboratory air (Bambi Oil Free Compressor, Model VTS75D).

Page 4, Paragraph 2: "It is true that the larger number of smaller particles does not necessarily compensate for the much larger volume of a smaller number of larger particles...."

In response to similar size concerns raised by both Reviewer #1 and #3 in their original reviews, additional measurements using smaller droplets show no difference in decay rate.

We feel that the Milton work (1) is the gold standard to measuring actual respiratory aerosol, as it does so directly from the mouth, i.e. it is a measurement at source rather than a measurement that is complicated by dispersion, particle loss and aging. As Reviewer #3 remarks, the work done by Kušan et al. (2) could be complicated by sedimentation losses due to factors such as the sampler being some distance away from the source, etc. This does not mean that larger drops are not there,

nor does it mean that they do not transmit the virus over short distances. In addition, there needs to be consideration of whether the sampling time was sufficient to allow robust quantification of the small number of larger particles. Reviewer #3's comment that "14 μm take 2 minutes to fall 1 m, rapidly removing them from indoor air" is not consistent with the re-evaluation of airborne transport by Prather *et al.* (3) and that anything smaller than 100 μm likely sediments much more slowly than a simple calculation would suggest due to advection on air currents in a room.

Also note that in the manuscript we already discussed the interplay between particle size and pH change due to changes in $[\text{CO}_2]$. From *Discussion: Risk of Transmission is Highly Affected by Ambient Concentrations of CO_2* : "The decay data measured in this study are for droplet sizes in the oral mode (initially $>50 \mu\text{m}$ diameter). We use the infectivity decay data from these large droplet measurements to inform estimates of transmission risk for the small aerosol fraction and to estimate the relative changes in risk that result from changes in $[\text{CO}_{2(\text{g})}]$. It should be noted, that the rate of pH reduction will be dependent on both droplet size and total acid content in the air, where smaller droplets will be neutralised at a faster rate. The effect of size was explored in this study to a limited degree, and was found to have a minimal effect within the size range explored (Supplemental Figure 6). The precise degree to which the droplet size affects the rate of neutralization (and subsequent increased viral aerostability) should be measured in the future."

Response to Reviewer #3's use of previously published data on the original strain of SARS-CoV-2 (Starts on Page 6, 3rd paragraph and goes onto page 7):

Reviewer #3 dedicates an entire page in their latest review to an estimate of the rate of viral infectivity loss in our system based upon assumptions collated from our previous publications (Figure D). There are multiple issues with how this critique has been constructed. In this analysis, the pH that a bulk sample of growth medium reaches after 24 hours is used to argue that the rate of decay we observe is too fast to be attributed to the pH change. This is akin to measuring the pH of a glass of water left out overnight (a macroscopic sample) to estimate the pH changes in an aerosol (a microscopic sample). However the pH dynamics in an aerosol droplet are vastly different to those in a bulk phase sample (we discussed this in our previous publication, Oswin *et al.*, 2022, doi.org/10.1073/pnas.2200109119, Section: *Droplet pH, Carbon Dioxide Partitioning, and the Rate of the Loss of Infectivity at High RH*). Furthermore, at no point do we argue that the aerosol pH is 9.5; this is simply being asserted by the reviewer. The decay rate used in the reviewer's analysis to estimate the half-life relates to the original (Wuhan) virus strain, which, as we have previously reported, has a lower pH sensitivity than the Delta variant being discussed here). For all these reasons, this line of argument, being used to propose that our findings are inconsistent, is flawed. We feel the interpretation of our data as presented in our manuscript is sound.

Page 7, Last sentence: "According to the current state of knowledge, it is an exaggeration to blame the climate-induced rise in CO_2 concentration for a change in the infectivity of airborne pathogens."

Discussion regarding the broader implications of our findings, and their relationship with changes in the $[\text{CO}_2]$ in the atmosphere, have been modified/toned down throughout the manuscript. These changes include:

Abstract, last line now reads:

These observations confirm the critical importance of ventilation and maintaining low CO₂ concentrations in indoor environments for mitigating disease transmission. Moreover, the correlation of increased CO₂ concentration with viral aerostability need to be better understood when considering the consequences of increases in ambient CO₂ levels in our atmosphere.

Discussion: Broader Implications: Paragraph 1 now reads:

This increase may be enough to improve viral transmission through both increasing the aerostability of the virus outdoors (Figure 2A), but also increasing the baseline [CO_{2(g)}] indoors as well. The degree to which [CO_{2(g)}] plays a role in disease transmission via changes in aerostability specifically needs to be explored further across a range of conditions and particle types.

Discussion: Broader Implications: Paragraph 2 now reads:

From the experimental and model data reported here, we hypothesize that the seasonality of respiratory viral infections at the populations level may be affected by indoor [CO_{2(g)}] as well as changes in RH. Further study is needed to explore this relationship across a broad range of respiratory viruses within respiratory particles of various sizes and compositions.

1. O. O. Adenaiye *et al.*, Infectious Severe Acute Respiratory Syndrome Coronavirus 2 (SARS-CoV-2) in Exhaled Aerosols and Efficacy of Masks During Early Mild Infection. *Clinical Infectious Diseases* **75**, e241-e248 (2021).
2. A. Cvitešić Kušan *et al.*, The size distribution of SARS-CoV-2 genetic material in airborne particles sampled in hospital and home care environments occupied by COVID-19 positive subjects. *Sci Total Environ* **892**, 164642 (2023).
3. J. Sills *et al.*, Airborne transmission of SARS-CoV-2. *Science* **370**, 303-304 (2020).

REVIEWERS' COMMENTS

Reviewer #1 (Remarks to the Author):

This is a revision of a previous manuscript I have reviewed. The revised manuscript satisfies my remaining comments, and I believe it is suitable for publication.